# Membrane-dependent actin polymerization mediated by the *Legionella pneumophila* effector protein MavH

Qing Zhang[1,2☯], Min Wan[1,2☯], Elena Kudryashova[3], Dmitri S. Kudryashov[3], Yuxin Mao[1,2]*

1 Weill Institute for Cell and Molecular Biology, Cornell University, Ithaca, New York, United States of America, 2 Department of Molecular Biology and Genetics, Cornell University, Ithaca, New York, United States of America, 3 Department of Chemistry and Biochemistry, The Ohio State University, Columbus, Ohio, United States of America

☯ These authors contributed equally to this work.
* ym253@cornell.edu

**Data Availability Statement:** All relevant data are within the manuscript and its Supporting Information files.

**Funding:** This work was supported by the National Institutes of Health (NIH) Grants R01-GM135379-

## Abstract

*L. pneumophila* propagates in eukaryotic cells within a specialized niche, the *Legionella*-containing vacuole (LCV). The infection process is controlled by over 330 effector proteins delivered through the type IV secretion system. In this study, we report that the *Legionella* MavH effector localizes to endosomes and remodels host actin cytoskeleton in a phosphatidylinositol 3-phosphate (PI(3)P) dependent manner when ectopically expressed. We show that MavH recruits host actin capping protein (CP) and actin to the endosome via its CP-interacting (CPI) motif and WH2-like actin-binding domain, respectively. In vitro assays revealed that MavH stimulates actin assembly on PI(3)P-containing liposomes causing their tubulation. In addition, the recruitment of CP by MavH negatively regulates F-actin density at the membrane. We further show that, in *L. pneumophila*-infected cells, MavH appears around the LCV at the very early stage of infection and facilitates bacterium entry into the host. Together, our results reveal a novel mechanism of membrane tubulation induced by membrane-dependent actin polymerization catalyzed by MavH that contributes to the early stage of *L. pneumophila* infection by regulating host actin dynamics.

## Author summary

*Legionella pneumophila*, a bacterium discovered in 1976, can cause a severe lung infection known as Legionnaires' disease. Upon inhalation of contaminated aerosol, *L. pneumophila* releases over 330 bacterial proteins to facilitate its intracellular survival and proliferation. These secreted bacterial proteins target a variety of host cellular pathways, including the actin cytoskeleton. In this study, we focused on one of such secreted *L. pneumophila* proteins, called MavH. We found that MavH promotes actin assembly both in vitro and in vivo on the membrane surface and actin polymerization triggered by MavH further drives the deformation and tubulation of the membrane. Moreover, we found that MavH promotes the entry of the bacterium into host cells by regulating host actin dynamics. In

01 (Y.M.) R01-GM114666 (D.S.K.). The funders had no role in study design, data collection and analysis, decision to publish, or preparation of the manuscript.

**Competing interests:** The authors have declared that no competing interests exist.

summary, we identified MavH as a novel actin regulator that induces actin polymerization-dependent membrane tubulation and enhances the ability of *L. pneumophila* to infect its host.

## Introduction

The gram-negative bacterium *Legionella pneumophila* is a facultative intracellular pathogen. Human infection, which occurs when aerosols contaminated by *Legionella* are inhaled, is found to be responsible for a severe form of pneumonia in humans known as Legionnaires' disease [1, 2]. *L. pneumophila* secretes over 330 effector proteins into host cells via its Dot/Icm (Defective Organelle Trafficking/Intracellular Multiplication) apparatus during infection [3–5]. These proteins modulate every step in the infection process, including host cell entry [6, 7], maturation of a replication-competent *Legionella* containing vacuole (LCV) [8], evading phagolysosomal fusion [9] and autophagy [10, 11], and final egress from the host cell [12]. Although the biological functions of many effectors have been elucidated, the exact molecular mechanisms of most effectors remain uncharacterized.

Actin is one of the most conserved proteins throughout eukaryotic evolution, whose cellular functions rely on the ability of actin monomers (G-actin) to polymerize into filaments (F-actin). F-actin is dynamically exchanged with G-actin with a net association of ATP-actin to the barbed (+) end and dissociation of ADP-actin monomers from the pointed (-) end [13]. The rapid assembly and disassembly of the actin cytoskeleton play an essential role in diverse cellular processes [14], including phagocytosis, micropinocytosis, endocytosis, vesicle trafficking, cell motility, polarity, and cytokinesis. The dynamics of the actin cytoskeleton are tightly regulated by a large number of actin-binding proteins, such as actin nucleators, capping proteins, severing proteins, bundling proteins, etc [13, 15, 16]. *De novo* F-actin assembly is facilitated by actin nucleators[17]. Three major classes of nucleators have been identified so far: the Arp2/3 complex [18]; the formins [19]; and the tandem actin-binding domain proteins, such as Spire [20] and Cobl [21]. The last promotes actin nucleation by bringing together several G-actin molecules upon their binding to tandem actin-binding Wiskott–Aldrich syndrome protein (WASP)-homology 2 (WH2) domains [22]. The dynamics of the actin cytoskeleton are channeled to a subset of nucleated filaments by the actin capping protein (CP), a heterodimer of structurally similar α- and β-subunits. CP binds to the barbed ends of actin filaments preventing filament elongation or shortening [23]. Extensive studies have revealed that CP participates in many cellular processes, including lamellipodia and filopodia formation [24] and regulation of endosomal trafficking by fine-tuning F-actin density around endosomes [25]. Importantly, the capping activity of CP is further regulated by multiple proteins that contain a conserved capping protein interaction (CPI) motif. These CPI motif-containing proteins recruit CP to specific cellular membrane locations [26] and/or allosterically inhibit the capping activity of CP [27].

Given the essential role of actin in cell physiology, many bacterial pathogens have evolved distinct strategies to target the host actin cytoskeleton to promote their survival, proliferation, and dissemination [28, 29]. Various extracellular pathogens deliver bacterial toxins and effectors to modify Rho family GTPases or actin itself through ADP-ribosylation, as well as other types of posttranslational modifications to disrupt host actin homeostasis and thus prevent pathogen uptake [30–33]. Intracellular bacterial pathogens, in general, establish more intimate and sophisticated relationships with their host by secreting numerous (often hundreds) toxins and effectors with a plethora of functions. Thus, they can secrete effectors mimicking actin

nucleators to promote host actin polymerization and facilitate host cell entry. For example, the virulence factor VopL from *V. parahaemolyticus*, like many other eukaryotic nucleators, dimerizes and promotes actin nucleation via its tandem WH2 domains [22, 34–36] and unique processive pointed-end elongation [37] via their tandem WH2 and C-terminal actin-binding domains.

*L. pneumophila* is a facultative intracellular parasite and not an exception to this rule. Recent studies have revealed that several *L. pneumophila* effectors target the host actin cytoskeleton. VipA is found as an actin nucleator with an unknown mechanism [35, 38]. By altering the host actin cytoskeleton, VipA interferes with host membrane trafficking and promotes the invasion of epithelial cells by filamentous *L. pneumophila* [38, 39]. The *L. pneumophila* effector, RavK is reported to disrupt actin structures by direct proteolytic cleavage of actin [40]. For two other *Legionella* effectors, LegK2 targets the Arp2/3 complex by phosphorylation, and WipA target Arp2/3 complex signaling pathway by dephosphorylation, to inhibit actin polymerization [41, 42]. Despite accumulating evidence, the exact mechanism and the biological significance of actin hijacking during *Legionella* infection are largely unknown.

In a screen to search for *Legionella* effectors that perturb host actin dynamics, we identified the *L. pneumophila* effector MavH which localizes to endosomes and promotes actin polymerization on the surface of the endosome. We found that the intact C-terminal phosphatidylinositol-3 phosphate (PI(3)P) binding domain of MavH is required for its endosomal localization and actin patch formation. We further showed that MavH has a CPI motif that recruits CP to the endosome. Our *in vitro* bulk and single-molecule actin polymerization assays revealed that MavH inhibits actin elongation in solution, however, it promotes robust membrane-associated actin assembly, driving membrane tubulation in PI(3)P-dependent manner. Interestingly, we showed that MavH localizes to the surface of the LCV at the very early stage of infection and facilitates bacterium entry into the host. Our study uncovered an intricate mechanism of membrane-bound actin polymerization coupled to tubulation of PI(3)P-containing membranes, catalyzed by a single-WH2-domain protein, *L. pneumophila* effector MavH.

## Results

### MavH induces F-actin patches around the endosome

Since actin is conserved in all eukaryotes and is a common target for a variety of pathogens, we performed a screen for *Legionella* effectors that perturb the host actin cytoskeleton. In this screen, 315 *Legionella* effectors were fused with an N-terminal GFP tag and transfected into HeLa cells. The cells were then stained with phalloidin for F-actin. In this screen, we identified MavH as a potential candidate that causes actin rearrangement. GFP-MavH exhibited a punctate localization and colocalized with the early endosomal marker EEA1 (Fig 1A and 1B) and the PI(3)P marker RFP-FYVE (S1A and S1B Fig) when exogenously expressed in HeLa cells. Interestingly, strong F-actin signals were observed on MavH-positive endosomes (Fig 1A and 1C), indicating that MavH may recruit F-actin to the endosomal surface. Moreover, exogenous expression of MavH also caused endosomal trafficking defects as evidenced by the delayed trafficking of epidermal growth factor (EGF) in cells transfected with MavH (S2 Fig).

MavH was previously shown to interact with PI(3)P lipids via its C-terminal lipid-binding domain [43]. Structure prediction with AlphaFold2 [44] revealed that the C-terminal domain (CTD) of MavH has a compact all-$\alpha$-helical fold (S3A and S3B Fig). To characterize the lipid-binding specificity of MavH, we performed liposome co-sedimentation assays and revealed that MavH binds preferentially to PI(3)P-containing liposomes (S3C and S3D Fig). We next mapped the key residues involved in PI(3)P binding. According to the structure, a pocket with positive electrostatic surface potentials is evident on the surface of the CTD, which is predicted

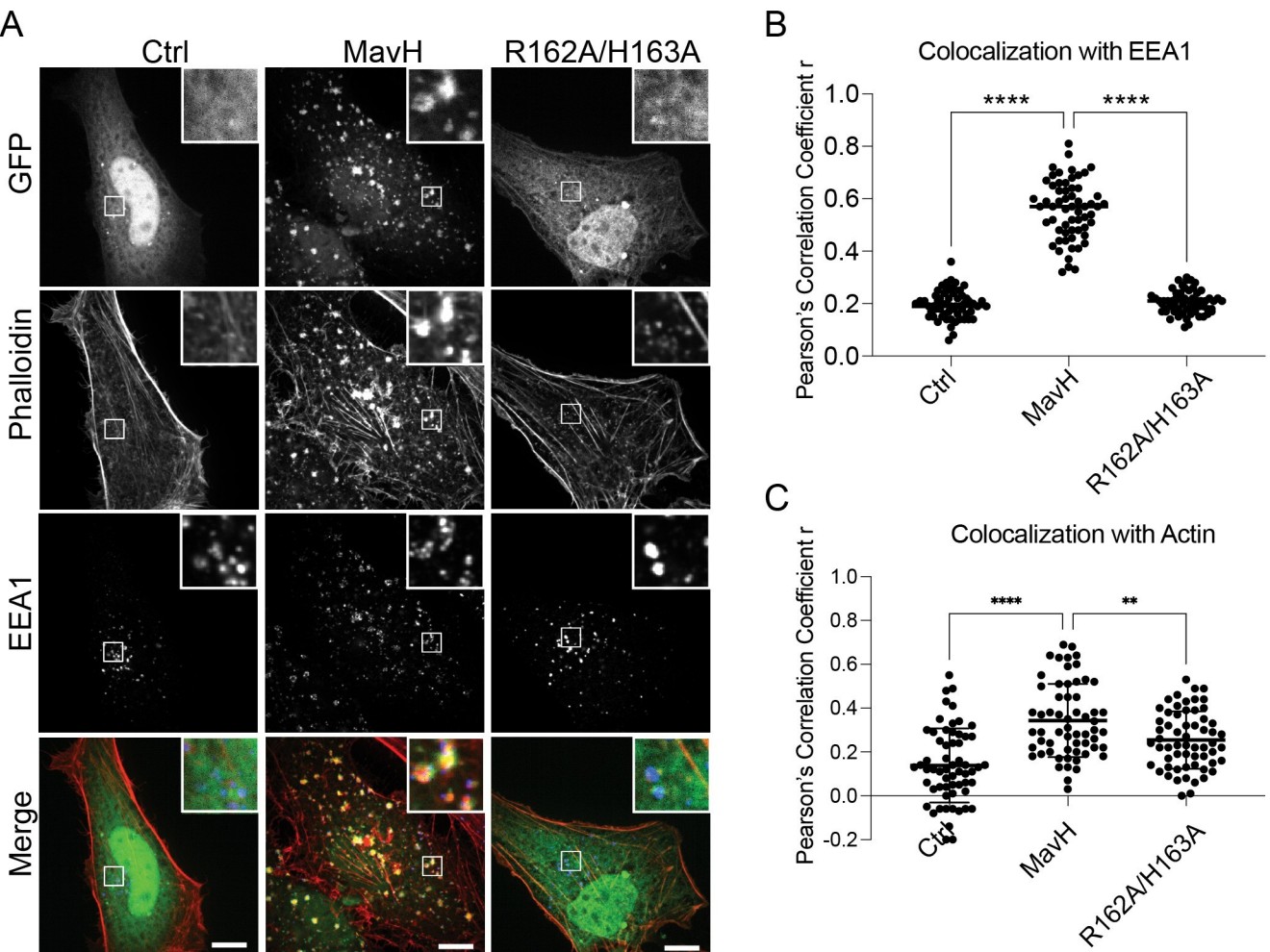

**Fig 1. MavH causes actin patch formation around endosomes.** (A) Localization of MavH in mammalian cells. HeLa cells were transfected with a plasmid expressing either GFP, GFP-MavH, or GFP-MavH R162A/H163A for 20 hours. Cells were fixed and immuno-stained with EEA1 antibodies followed by Rhodamine-conjugated phalloidin and then imaged by confocal microscopy. Scale bars, 10 μm. (B) (C) Colocalization of GFP fusion proteins with EEA1 or actin. Colocalization was determined by Pearson's correlation coefficient r. Data are depicted as scatterplots showing mean ±SD from 60 cells in 3 independent experiments. Statistical significance was assessed using one-way ANOVA (****$P<0.0001$, ***$P<0.001$, **$P<0.01$).

to mediate PI(3)P binding (S3E and S3F Fig). Indeed, PI(3)P-binding was substantially impaired by the MavH R162A/H163A mutant, which was designed to disrupt the positive charges at the predicted PI(3)P- binding pocket (S3G and S3H Fig). In agreement with the *in vitro* assays, MavH R162A/H163A mutant exhibited a cytosolic localization, and no endosomal actin patches were observed in cells expressing this mutant (Fig 1A–1C). Together, these results suggest that the PI(3)P-binding CTD is required for MavH endosomal localization and actin patch formation around the MavH-positive endosomes.

## MavH interacts with capping protein (CP) via a conserved CPI motif

To elucidate the molecular mechanism of MavH in actin polymerization, we performed a sequence analysis of MavH using HHpred (https://toolkit.tuebingen.mpg.de/tools/hhpred). We found that the central region of MavH contains a conserved sequence stretch that resembles the consensus sequence LxHxTxxRPK(6x)P characteristic for the capping protein

interaction (CPI) motif (Fig 2A). The CPI peptide wraps around the stalk region of the mushroom-shaped CP complex and targets the CP to specific cellular membrane locations to regulate the dynamics of the actin cytoskeleton [26, 45]. To test whether MavH has a functional CPI motif, we first co-expressed the CP complex (HA-tagged α subunit and mCherry-tagged β subunit) with GFP-MavH in HEK-293T cells to analyze the recruitment of CP by MavH. CP showed a diffused cytosolic localization in cells expressing GFP control whereas it colocalized with MavH to punctate structures in cells expressing wild-type GFP-MavH. Accordingly, the MavH CPI motif mutant, GFP-MavH R73A/K75A failed to recruit CP to punctate structures when overexpressed in cells (Fig 2B). Furthermore, the colocalization between MavH and CP to punctate structures was detected in cells expressing a MavH truncation mutant (MavH Δ53), of which the N-terminal 53 residues were deleted, but not in cells expressing the MavHΔ93 mutant, which in addition lacks the CPI motif (Fig 2B). Next, we assessed the direct interaction between MavH and CP. GFP-tagged MavH and the CP complex were co-transfected in HEK-293T cells and cell lysates were prepared after transfection for 2 days. GFP-MavH was immunoprecipitated by the resin conjugated with anti-GFP nanobodies and the CP α subunit was detected from fractions co-immunoprecipitated with wild-type MavH but not its CPI mutant (MavH R73A/K75A) (Fig 2C). The interaction between MavH and CP appears to be direct as evidenced by the pull-down of purified CP complex with the immobilized recombinant MavH protein (Fig 2D). These results suggest that the MavH CPI motif is required to mediate the interaction with CP and intracellular recruitment of CP to punctate structures.

We then asked whether the CPI motif of MavH is also responsible for the actin patch formation at endosomes. To test this, we transfected GFP-tagged MavH wild-type and mutants into HeLa cells and stained the cells with phalloidin. Interestingly, the CPI motif mutant, MavH R73A/K75A still induced actin patches around the endosome comparable to wild-type MavH (Fig 2E). However, no significant actin patch was detected in cells expressing MavH Δ93 or even MavH Δ53, which has an intact CPI motif and can recruit the CP to endosomes (Fig 2E). These results suggest that the N-terminal region, but not the CPI motif, is responsible for actin recruitment to the endosome.

## MavH interacts with actin via an N-terminal WH2-like domain

To address how MavH promotes actin assembly, we analyzed the primary sequence of the N-terminal region of MavH. Multiple sequence alignment revealed a cluster of several conserved hydrophobic residues at the beginning of the predicted N-terminal α helix (Fig 3A and 3B). This structural feature is reminiscent of the actin-binding WH2 domain, which consists of one α- helix with few exposed hydrophobic residues engaging in a hydrophobic cleft formed between the subdomains 1 and 3 on actin [46]. To test whether the N-terminal α-helix mediates the interaction with actin, we expressed GFP-tagged wild-type MavH, the PI(3)P-binding domain mutant (MavH R162A/H163A), and the tentative WH2 domain mutant (MavH V24D/L31D) in HEK293T cells and assessed the interaction between MavH and actin by co-immunoprecipitation. Indeed, wild-type MavH, as well as its PI(3)P-binding mutant, was able to pull down actin, while the interaction with actin was substantially impaired by MavH-V24D/L31D mutant(Fig 3C). The interaction between actin and MavH was further mapped to the N-terminal region containing the predicted first α helix, as evidenced by the pull-down of actin by MavH construct consisting of residues 13–65 but not the MavH 13–65 V24D/L31D mutant (Fig 3C). These data suggest that MavH harbors an N-terminal α-helix that resembles a WH2 domain and mediates actin binding.

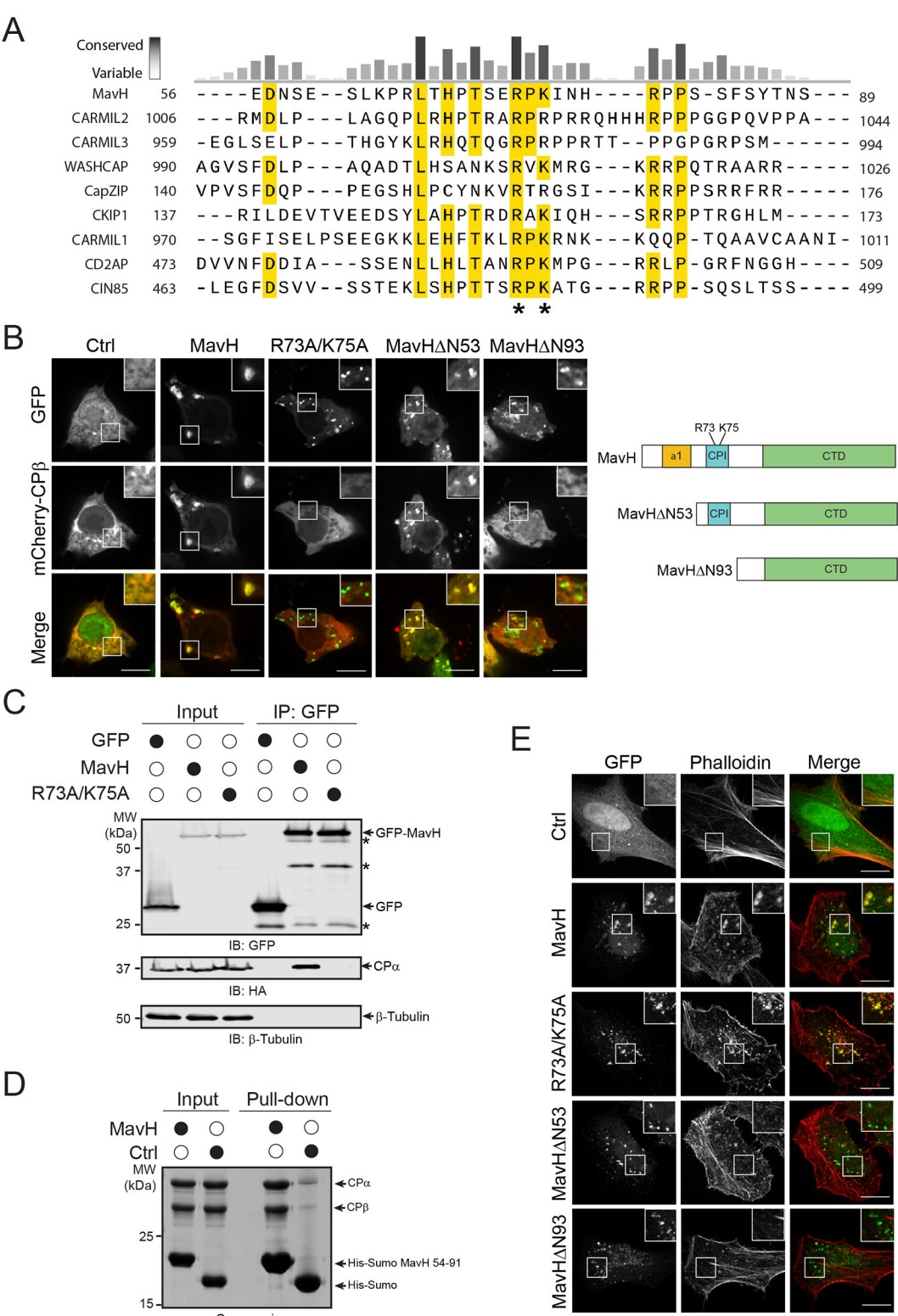

**Fig 2. MavH recruits capping protein (CP) via a conserved CPI motif.** (A) Multiple sequence alignment of MavH with the CPI motif family. The sequences corresponding to the CPI motif were aligned by Clustal Omega. Identical residues and similar residues are highlighted in yellow. Two conserved positive charged residues (R73 and K75) are highlighted with "*". Uniprot accession numbers for MavH: Q5ZSU1; CARMIL2: Q6F5E8; CARMIL3: Q8ND23; WASHCAP: Q9Y4E1; CapZIP: Q6JBY9; CKIP1: Q53GL0; CARMIL1: Q5VZK9-1; CD2AP: Q9Y5K6; CIN85: Q96B97. (B) Recruitment of CP by MavH is

dependent on the CPI motif. GFP-tagged MavH constructs were co-expressed with mCherry-CPβ-HA-CPα in HEK293T cells. Cells were fixed and imaged by confocal microscopy. Scale bars, 10 μm. (C) Co-immunoprecipitation of GFP-tagged MavH proteins with CP. HEK293T cells were co-transfected with mCherry-CPβ-HA-CPα with GFP empty vector or GFP-tagged MavH or GFP-tagged MavH R73A/K75A. GFP-tagged proteins were immunoprecipitated from whole-cell lysates with anti-GFP antibodies and then analyzed by SDS-PAGE followed by immunoblot with both anti-HA and anti-GFP antibodies. (D) In vitro pull-down assay of CP by His-Sumo or His-Sumo-tagged MavH. Purified His-Sumo or His-Sumo-tagged MavH (a.a. 54–91) as bait proteins were loaded onto cobalt beads to pull down recombinant CP. Pull-down materials were resolved by SDS-PAGE and stained by Coomassie-blue dye. (E) The rearrangement of the actin cytoskeleton around endosomes by MavH is independent of the CPI motif. HeLa cells were transfected with GFP-tagged MavH wild-type, CPI motif mutant or truncations, and stained for actin with Rhodamine conjugated phalloidin. Scale bars, 10 μm.

To investigate the effects of the N-terminal WH2-like domain of MavH on actin dynamics, we overexpressed GFP-tagged MavH and its mutants in HeLa cells and examined their localization and actin structures in transfected cells. Strikingly, although MavH-V24D/L31D showed a punctate localization, no actin signals were detected on the MavH mutant positive puncta in contrast to wild-type MavH (Fig 3D). In agreement with the co-IP assay, the N-terminal WH2-like domain alone was sufficient to target GFP-MavH-WH2-13-65 to the actin cytoskeleton, whereas the V24D/L31D mutant counterpart was completely cytosolic (Fig 3D).

To further validate the role of the N-terminal WH2-like domain, we fused the N-terminal region 13–65 of MavH with another PI(3)P-binding domain from the *Legionella* effector SetA (SetA-CTD), which is localized to endosomes when expressed in eukaryotic cells [47, 48] but does not recruit actin (S4 Fig). Similar to wild-type MavH, the chimeric protein (MavH-SetA) also exhibits an endosomal localization and, in contrast to the parental SetA-CTD, induces actin patch formation around the endosomes (S4 Fig). These results support that the N-terminal portion of MavH is responsible for actin recruitment to the endosome compartment.

MavH intracellular localization and MavH-mediated actin polymerization on intracellular membrane-bound organelles were further recapitulated in the budding yeast, *Saccharomyces cerevisiae* (S5 Fig). We transformed GFP- or mCherry-tagged MavH constructs in yeast cells that were under the control of a galactose-inducible promoter. Wild-type MavH, as well as MavH constructs that have an intact PI(3)P-binding CTD, was found to be enriched on the surface of yeast vacuoles labeled with mCherry-tagged Vph1 marker. In contrast, the PI(3)P-binding mutant (MavH-R162A/H163A) and the N-terminal WH2 region of MavH (MavH-13-65) showed a peripheral punctate localization (S5A Fig). These results suggest that the localization of MavH to the vacuole is dependent on its C-terminal PI(3)P-binding domain. Like in mammalian cells, wild-type MavH and its CPI mutant (MavH-R73A/K75A) induced robust actin polymerization on the surface of the vacuole. However, MavH mutants, which are either incapable of actin-binding (MavH V24D/L31D and MavH CTD-120-266) or defective in membrane binding (MavH-R162A/H163A and MavH-WH2-13-65), failed to polymerize actin on the vacuole (S5B Fig). Furthermore, MavH constructs containing the actin-binding domain but deficient in membrane-binding region (MavH-R162A/H163A and MavH-13-65) displayed colocalization with peripheral actin patches, consistent with the interaction of MavH WH2-like domain with actin. Interestingly, over-producing MavH proteins that contain the intact WH2-like domain is toxic to yeast cells (S5C Fig), indicating that the WH2-like domain causes yeast growth defects likely by interfering with endogenous actin dynamics.

In summary, our results revealed that the N-terminal region of MavH harbors an actin-interacting motif. This WH2-like domain, together with the C-terminal PI(3)P-binding domain, is responsible for actin assembly on the surface of endosomes.

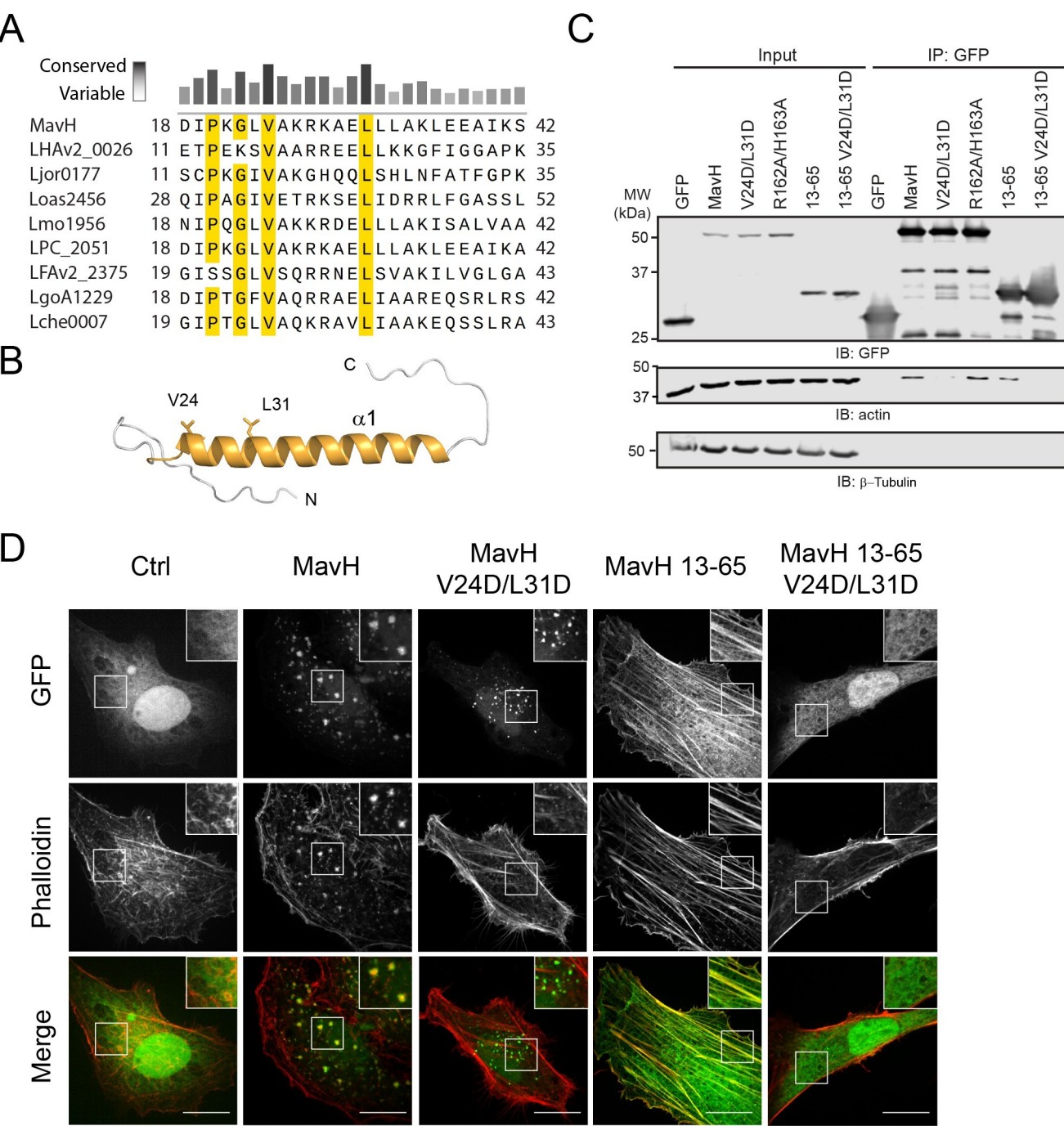

**Fig 3. MavH contains an N-terminal WH2-like domain and interacts with actin.** (A) Multiple sequence alignment of N-terminus of MavH with other *Legionella* homologs. Identical and similar residues are highlighted in yellow. Of note, highly conserved residues are all hydrophobic. (B) Predicted structure of MavH N-terminus with AlphaFold2. Two highly conserved hydrophobic residues (V24D and L31D) are shown in sticks. (C) MavH interacts with actin via the N-terminus. HEK-293T cells were transfected with GFP empty vector or GFP-tagged MavH wild-type, truncations, and mutants. GFP-tagged proteins were immunoprecipitated from whole-cell lysates with anti-GFP antibodies and then analyzed by immunoblot with both anti-actin and anti-GFP antibodies. (D) The rearrangement of the actin cytoskeleton caused by MavH is dependent on interaction with actin via the N-terminus. HeLa cells were transfected with GFP-tagged wild-type MavH, mutants, or truncations and stained for actin with Rhodamine conjugated phalloidin. Scale bars, 10 μm.

## MavH mediates membrane-dependent actin polymerization

To elucidate the molecular mechanism of actin assembly catalyzed by MavH, we performed in vitro pyrene-actin polymerization assays [49]. To our surprise, wild-type MavH did not promote actin assembly in F-actin buffer, instead, it inhibited actin polymerization compared to the actin alone control (Fig 4A). A similar inhibitory effect was also observed for the lipid-binding motif mutant (R162A/H163A). In contrast, the MavH WH2 mutant (V24D/L31D) showed no effect on actin polymerization (Fig 4A).

Since MavH promoted actin assembly on endosomes in the cell, we reasoned that MavH-triggered actin assembly may require the membrane. To test this idea, we performed the pyrene-actin assays in the presence of PI(3)P-containing liposomes (PC: PS: PI(3)P = 8: 1: 1). Liposomes alone did not affect actin polymerization, however, in the presence of both liposomes and wild- type MavH, actin polymerization was enhanced, particularly at the initial stage (Fig 4B), which is typical for actin nucleating proteins. Unfortunately, the extend of the enhancement was impossible to assess accurately, as the fluorescence signals exhibited an abnormal fluctuation, often observed when the light from a fluorophore is scattered by large particles (e.g., filament bundles). To test this hypothesis, we performed similar actin polymerization assays and visualized the final products by confocal microscopy following the staining with Alexa Fluor 488-phalloidin. Strikingly, a massive accumulation of F-actin was observed congregating around the DiI-labeled PI(3)P-containing liposomes, concomitant with liposome deformation and clustering (S6 Fig). As a control, the MavH-V24D/L31D showed no effect on actin polymerization (Fig 4B), and no significant F-actin signal was observed around the liposomes (S6 Fig).

The importance of membrane binding in MavH-mediated actin polymerization was further validated by in vitro pyrene-actin assays when the membrane association of MavH was perturbed. The PI(3)P-binding defective mutant, MavH-R162A/H163A showed no stimulation of actin polymerization (Fig 4B), and no F-actin was detected around the liposomes (S6 Figs). Along this line, liposomes lacking PI(3)P also failed to promote actin polymerization triggered by wild-type MavH (Figs 4B and S6). Together, these data suggest that although MavH contains a single actin-binding WH2-like domain and inhibits actin polymerization in solution, it promotes F-actin assembly on PI(3)P-containing membranes upon its association with the membrane via its C-terminal PI(3)P-binding domain.

To further elucidate the effects of MavH on actin polymerization at the single-filament level, we performed an in vitro reconstituted total internal reflection fluorescence microscopy (TIRFM) analysis. In the presence of MavH, actin filaments were much shorter than the control "actin only" filaments (Fig 4C and 4D) corroborating the results from the bulk pyrene-actin assays (Fig 4A and 4B). Plotting filament length versus time revealed multiple long pauses in the growth of filaments elongated in the presence of MavH regardless of the PI(3)P presence (Fig 4E) implying that actin elongation is severely inhibited by MavH. Actin nucleation measured as the number of filaments present in a field of view at a 4-min time point was moderately increased by MavH and further enhanced when the reaction was also supplemented with the liposomes (Fig 4F). In the absence of PI(3)P, MavH-nucleated filaments did not associate with the membranes. In striking contrast, MavH assembled actin mainly on the surface of the PI(3)P-containing liposomes: actin filaments appeared to nucleate, grow and bundle at the membrane, which precluded measurements of actin nucleation. Many PI(3)P-containing liposomes surrounded by MavH-assembled actin appeared to have protrusions and/or were elongated (Fig 4C). Therefore, although MavH per se inhibits actin elongation in solution, the net effect in the presence of PI(3)P-containing membrane is the robust actin assembly, which can deform/remodel the membrane.

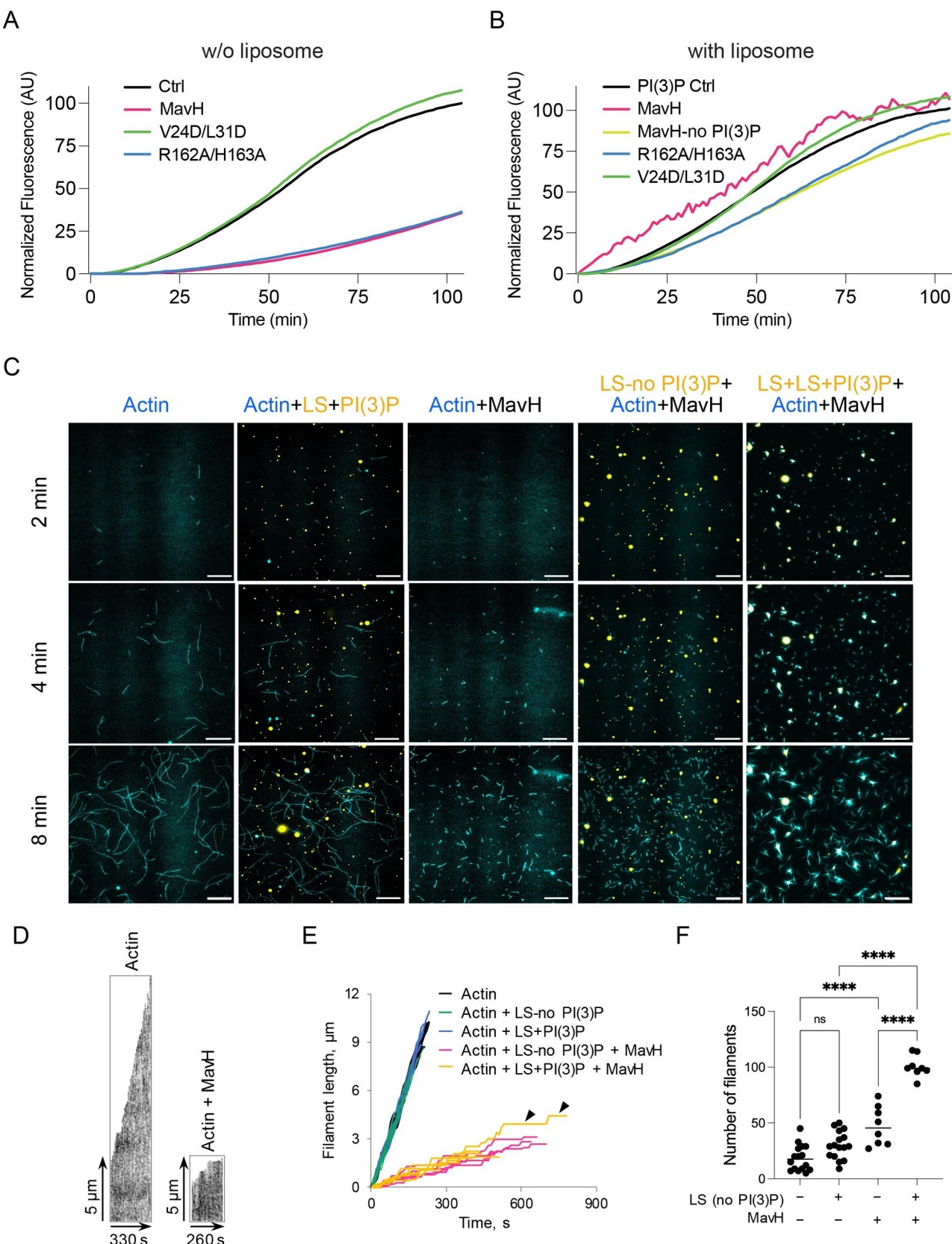

**Fig 4. MavH inhibits actin elongation but promotes actin assembly on PI(3)P-positive liposomes.** Pyrene-actin polymerization assays of MavH without (A) and with (B) PI(3)P-containing liposomes. All reactions contain 3 μM actin (10% pyrene-actin) and 250 nM wild-type or mutant MavH proteins. Actin polymerization was initiated by adding 10X actin polymerization buffer and fluorescence signals (arbitrary units (AU)) were recorded over time. (C-F) In vitro TIRFM reconstituted assays of actin polymerization were conducted as described in Methods in the absence or presence of 250 nM MavH and 50 μM liposomes (LS). (C) Representative time-lapsed TIRFM images of Alexa Fluor 488-labeled actin and DiD-labeled liposomes are shown at three-time points. Scale bars are 10 μm. (D) Comparison of elongation of representative actin filaments growing in the absence or presence of MavH was assessed using kymographs. (E) Filament elongation was measured by plotting individual filament lengths versus time. Note long pauses in the filament growth in the presence of MavH (denoted by arrowheads) indicative of actin barbed-end leaky capping activity of MavH. (F) Actin nucleation was assessed by counting the number of filaments at a 4-min time point. Data pooled from 2–3 independent experiments each containing four separate fields of view are shown as dots, means are represented as horizontal lines. Statistical significance was determined using ordinary one-way ANOVA with multiple comparisons (****$p < 0.0001$; ns, non-significant ($p = 0.0576$)).

## MavH-mediated actin polymerization induces membrane tubulation

In diverse cellular processes, the actin network has been implicated in membrane deformation by generating pulling or pushing forces[50]. To examine morphological changes of liposomes caused by membrane-dependent actin polymerization mediated by MavH, we employed TIRFM and electron microscopy (EM) approaches. Strikingly, lipid protrusions/tubules were induced along the MavH-polymerized actin filaments on PI(3)P-containing liposomes as revealed by time-lapse TIRFM imaging (Fig 5A and S1 Video). In contrast, in the absence of PI(3)P, F-actin filaments were formed independent of liposomes and the liposomes remained intact (Fig 5B and S2 Video). Importantly, the PI(3)P-liposome protrusions mediated by MavH were growing with rates similar to those observed for slowly growing individual actin filaments in the presence of MavH (Fig 5C), suggesting a direct coupling between the two events. Membrane tubulation induced by MavH-mediated actin polymerization was also confirmed by transmission EM, which further revealed that the protruded membrane tubules were decorated with longitudinal F-actin fibers (Fig 5D and 5E). Together, our TIRFM and EM imaging studies demonstrate that MavH-mediated actin polymerization induces membrane deformation/tubulation in a PI(3)P-dependent manner.

## MavH recruits CP to negatively regulate actin polymerization at the membrane

We next asked about the role of the CPI motif in MavH-mediated actin assembly. It has been reported that capping protein regulates F-actin density around endocytic vesicles [25, 51]. We speculated that MavH might recruit CP to regulate F-actin density at PI(3)P-containing membranes. To test this hypothesis, we first generated a CP mutant (CPβ-R15A), which carries R15A substitution at its β-subunit. This mutant is defective in binding with the CPI motif (Fig 6A) but retains normal actin-capping activity [26]. As expected, this mutant failed to be recruited to the liposome by MavH (Fig 6B) but exhibited a comparable capping activity to wild-type CP as evidenced by a similar inhibitory effect on actin polymerization (Fig 6C and 6D). We then analyzed the effect of the wild type and the mutant CP on MavH-catalyzed actin polymerization. We observed that wild-type CP substantially inhibited MavH-mediated actin polymerization while the CPβ-R15A mutant displayed a milder inhibition on actin polymerization (Fig 6D). Correspondingly, the F-actin signal around the liposome was substantially weaker and the liposomes were less aggregated in the presence of wild-type CP compared to that of CPβ-R15A mutant (Fig 6E). Together, these data suggest that the recruitment of CP via the CPI motif negatively regulates actin polymerization.

## MavH localizes to the LCV membrane and promotes actin assembly at the early stage of *Legionella* infection

We next examined the intracellular localization of MavH during intracellular infection by *L. pneumophila*. We first created a MavH deletion strain and strains supplemented with a

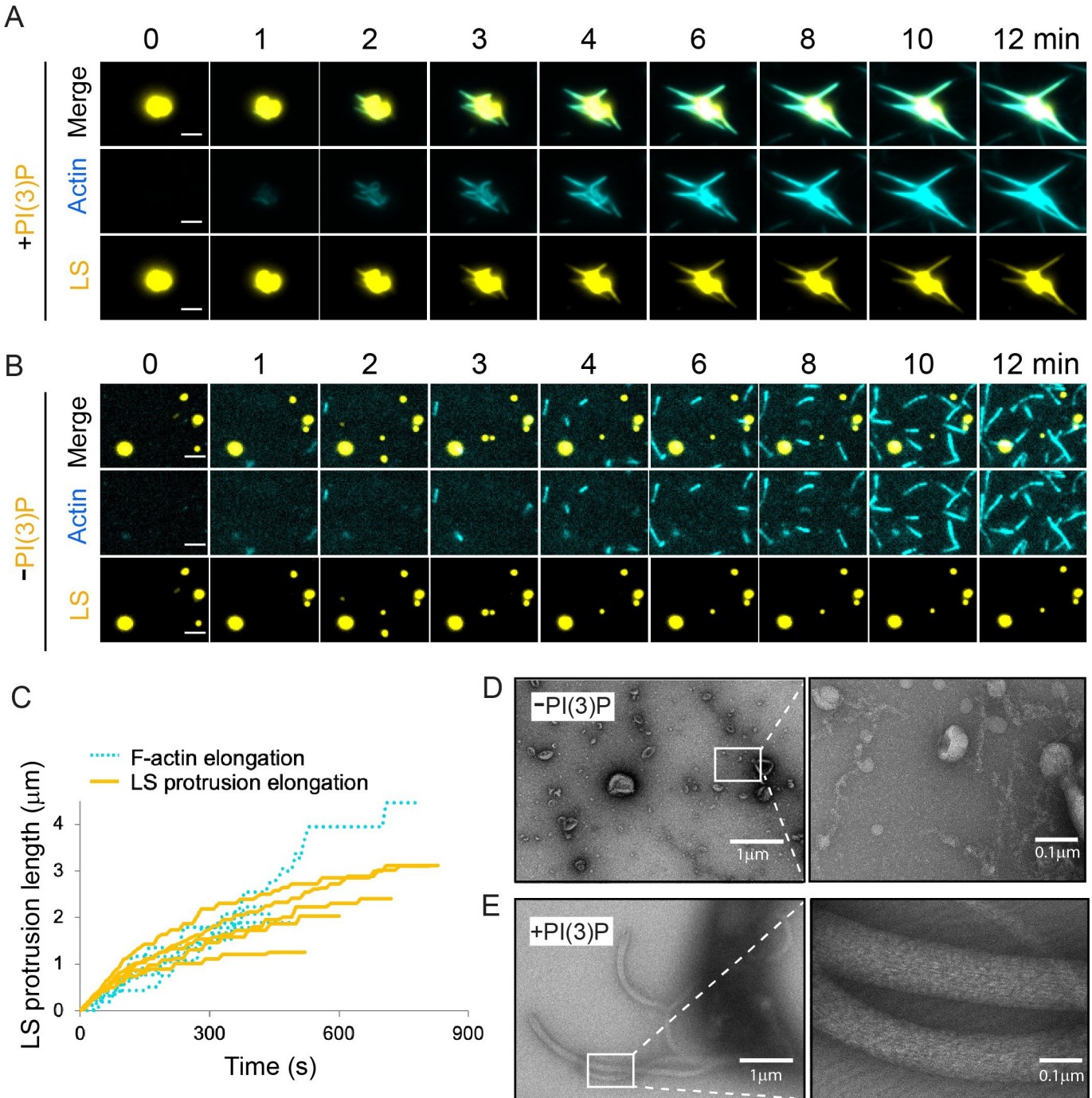

**Fig 5. Actin polymerization induced by MavH drives membrane tubulation.** (A, B) In vitro TIRFM reconstituted assays of actin polymerization were conducted as described in Methods in the presence of 250 nM MavH and 50 μM of either PI(3)P-containing liposomes (A) or liposomes without PI(3)P (B). Representative time-lapsed TIRFM images of Alexa Fluor 488-labeled actin and DiD-labeled liposomes are shown. Scale bars are 2.5 μm. See also S1 and S2 Videos. (C) Length of PI(3)P-liposome protrusions mediated by MavH-induced actin assembly (as shown in A) was plotted over time and superimposed with the F-actin elongation rates shown in Fig 4F (Actin + LS(+PI(3)P) + MavH). (D, E) EM images of negatively stained liposomes incubated with actin and MavH after induction of actin assembly. Images were taken at a magnification of 5,300X (left) and 92,000X (right). Reactions were performed using 6 μM actin, 500 μM PI(3) P-containing liposomes, and 1 μM MavH.

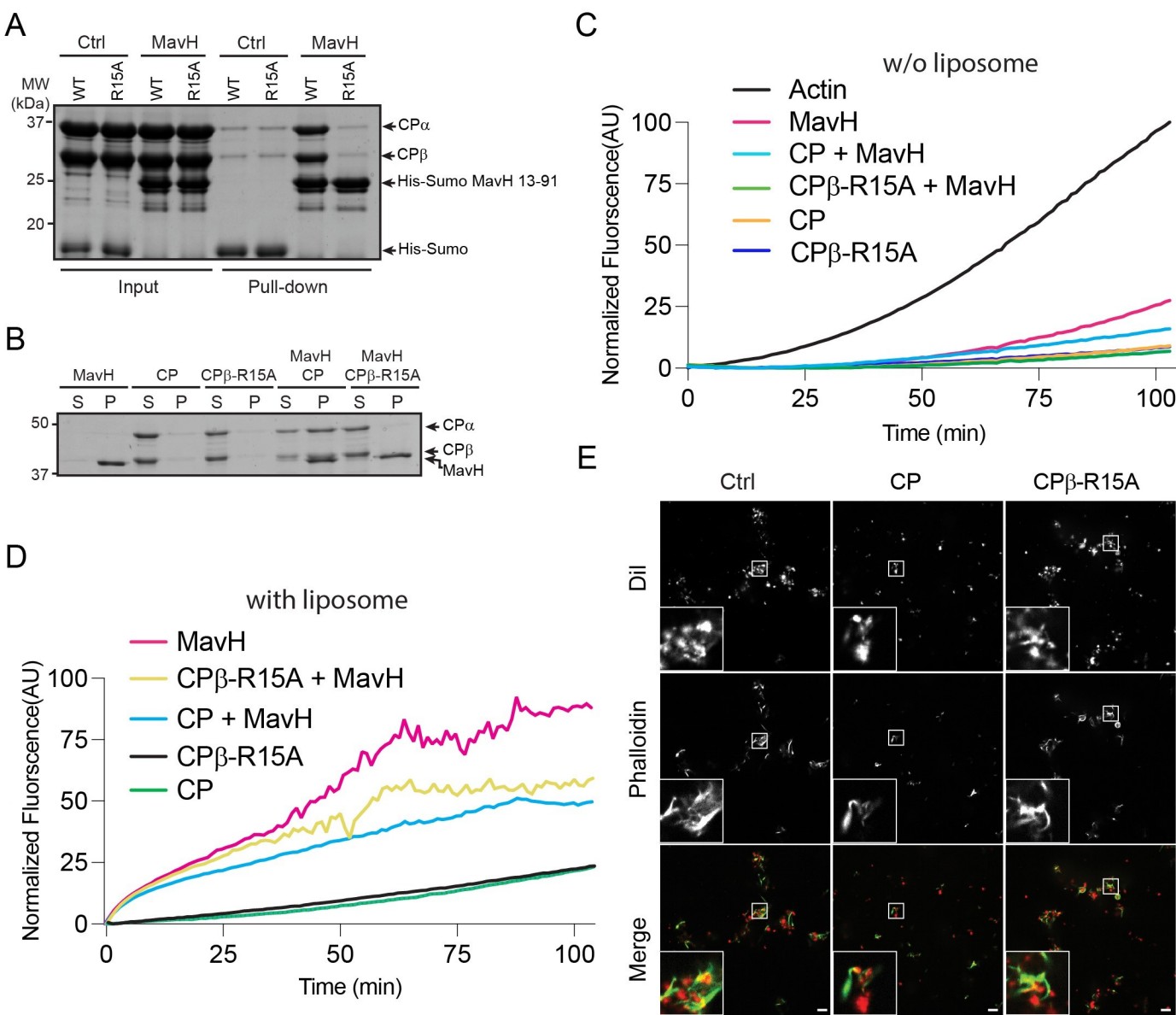

**Fig 6. The CPI motif of MavH recruits CP and negatively regulates actin polymerization.** (A) In vitro pull-down assay of CP by MavH. Cobalt beads preloaded with His-Sumo or His-Sumo-tagged MavH (a.a. 13–91) were used as the bait to pull down purified wild-type CP or CPβ-R15A. Pull-down materials were resolved by SDS-PAGE and stained by Coomassie-blue dye. (B) Liposome co-sedimentation assay of CP or CPβ-R15A with MavH. Purified proteins were incubated with PI (3)P-containing liposomes and then spun down by ultracentrifugation. The supernatant (S) and pellet (P) fractions were analyzed by SDS-PAGE followed by Coomassie-blue staining. (C) Pyrene-actin polymerization assays with actin alone or in the presence of CP, CPβ-R15A, and MavH. (D) Pyrene actin polymerization assay of CP, CPβ-R15A, and MavH in the presence of PI(3)P-containing liposomes. (E) In vitro liposome imaging assay. Reactions were performed using 3 μM actin, 250 nM MavH, and either 25 nM CP or CPβ-R15A mutant. PI(3)P-containing liposomes were used at 250 μM. After 30 min incubation, actin was stained with Alexa Fluor 488-phalloidin, and the reaction products were imaged with fluorescence confocal microscopy. Scale bars, 10 μm.

plasmid expressing wild-type or mutant MavH fused with an N-terminal 4xHA tag. These strains were then used to infect HEK293T cells expressing FcγRII receptor. After infection for 10 min, cells were fixed with ice-cold methanol and immunostained with an anti-HA antibody. HA signals were detected around the LCV in cells infected with WTΔ*mavH* supplemented with 4xHA-MavH but not with Δ*dotA* strain also overexpressing 4xHA-MavH but defective of effector translocation (Fig 7A). We then inspected the time course of the retention

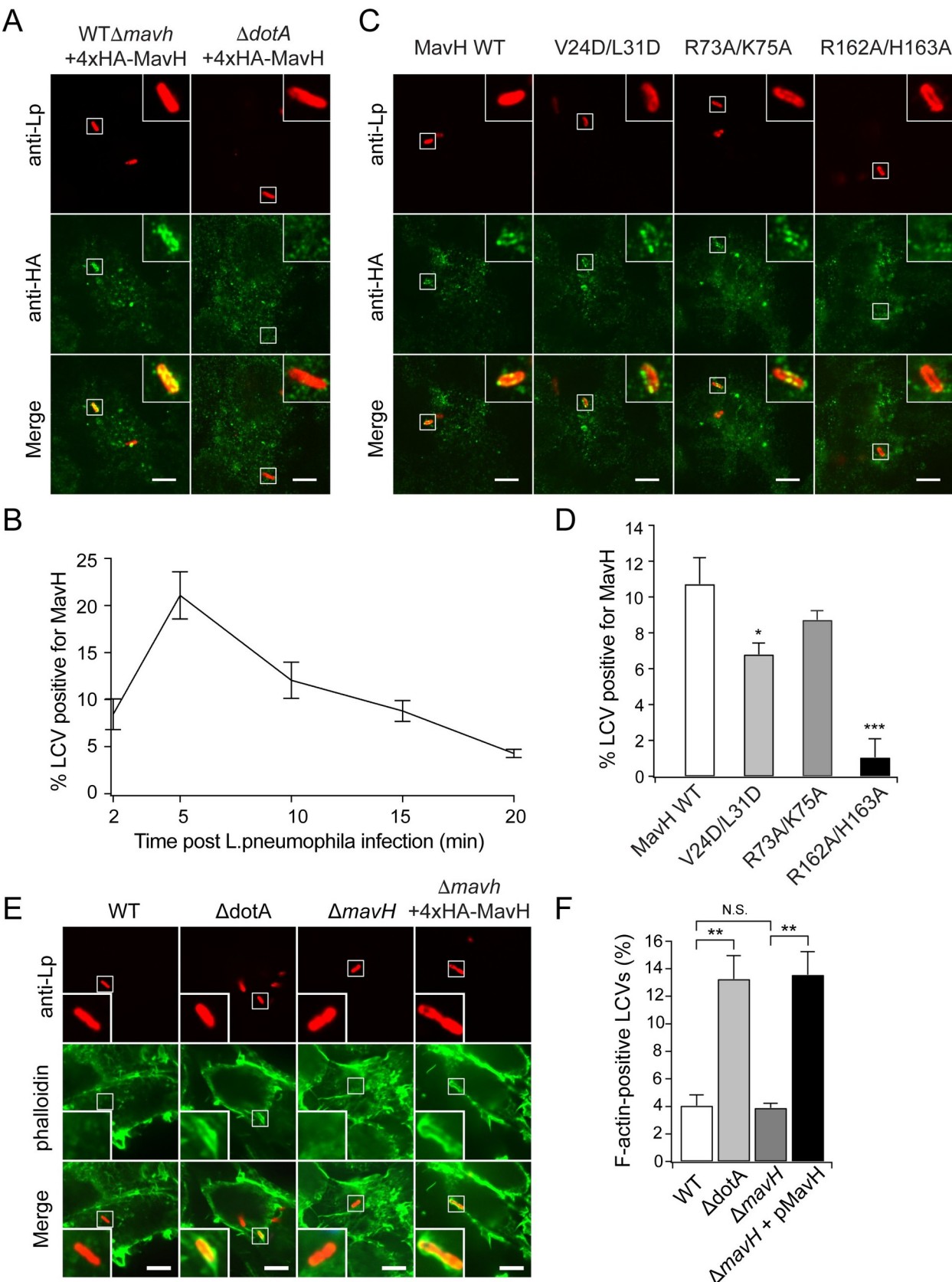

**Fig 7. MavH localizes to the surface of LCV and facilitates LCV accumulation of F-actin at the early stage of *Legionella* infection in HEK-293T cells.** (A) Representative images show localization of 4xHA-MavH at 5 min post-infection. FcγRII-expressing HEK293T were challenged by wild-type Δ*mavH* or Δ*dotA* strains supplemented with a plasmid expressing 4xHA-MavH for 5 min. Cells were fixed using 4% PFA, for 15 min and then permeabilized using ice-cold methanol for 10 min. 4xHA-MavH was immunostained using mouse-anti-HA primary antibodies and Alexa-488 anti-mouse secondary antibodies. Representative images showed localization of 4xHA-MavH at 5 min post-infection. Scale bars are 10 $\mu$m. (B) Quantifications of LCVs positive for 4xHA-MavH in HEK-293T cells infected by WTΔ*mavH* overexpressing 4xHA-MavH for the indicated time, shown as mean ± SEM from three independent experiments. At least 40 LCVs were analyzed per condition. *P<0.05 and ***P<0.001. (C) FcγRII-expressing HEK293T cells were infected by *mavH* deletion strain overexpressing 4xHA-MavH wild-type or indicated mutant for 10 min. cells were fixed and immunostained with anti-HA antibodies as in (A). Scale bars are 10 $\mu$m. (D) Quantifications of MavH-positive LCVs. The percentage of MavH-positive LCVs was shown as mean ± SEM from three independent experiments. At least 40 LCVs were analyzed for condition. *P<0.05 and ***P<0.001. (E) Representative images show the localizations of *Legionella* bacteria (red) and Alexa Fluor 488-phalloidin-labeled F-actin (green) in FcγRII-expressing HEK-293T challenged by the indicated *Legionella* strains for 10 min. Scale bars are 10 $\mu$m. (F) Quantification of LCVs positive for F-actin in HEK-293T cells in (E) shown as mean ± SEM from three independent experiments. At least 50 LCVs were analyzed for each time point. **P < 0.01, N.S. = not significant.

of MavH at the LCV. MavH was detected at the LCV as early as 2 min p.i. and peaked at around 5 min p.i. (~ 20% LCVs are positive for MavH). MavH signals were gradually reduced as the infection progressed (Fig 7B). Similar results were observed in the infection of differentiated human macrophage U937 cells (S7A and S7B Fig). We further investigated the functional determinants for the anchoring of MavH to the LCV. We observed that the WH2 actin-binding mutant, MavH-V24D/L31D showed a slight reduction of LCV localization, while the CPI mutant, MavH-R73A/K75A, exhibited no discernable difference compared to the wild-type MavH. However, nearly no HA signals could be detected at the LCV for the PI(3)P-binding mutant, MavH-R162A/H163A (Fig 7C and 7D). Together, these results demonstrated that MavH localizes to the LCV at the early stage of *Legionella* infection and the anchoring of MavH to the LCV requires its C-terminal PI(3)P-binding domain.

We next analyzed the effect of MavH on the actin cytoskeleton during *Legionella* infection. HEK-293T cells expressing FcγRII receptor were infected with wild-type and mutant *Legionella* strains for 10 min and the percentage of LCVs positive for F-actin was quantified by fluorescence microscopy. In agreement with previously reported results [41], a very low percentage of LCVs were positive for F-actin (~4%) when cells were infected with the wild-type strain and more than 3-fold higher F-actin positive LCVs were detected in cells challenged with the Δ*dotA* mutant strain. Although no discernable change in the percentage of F-actin-positive LCVs was detected when cells were infected with the Δ*mavH* strain compared to the wild-type bacteria, the percentage of F-actin-positive LCVs was substantially increased in cells infected with the strain overexpressing MavH (Fig 7E and 7F). Similar results were also observed in the infection of differentiated U937 cells (S7C and S7D Fig). Together, our results suggested that MavH promotes actin assembly at the LCV at the early stage of infection contributing to the intricate mechanisms involved by *Legionella* to hijack host actin cytoskeleton (discussed below).

## MavH facilitates the entry of *L. pneumophila* into the host

We next investigated the role of MavH in *L. pneumophila* intracellular proliferation. The intracellular growth of the MavH deletion strain and the strain supplemented with a plasmid expressing MavH showed no obvious defects in *Acanthamoeba castellanii* compared to that of the wild-type strain (Fig 8A), indicating MavH is dispensable for intracellular growth in this organism. However, given that MavH promotes F-actin assembly at the early stage of bacterial infection and actin polymerization plays an important role in phagocytosis, we asked whether MavH facilitates bacterial uptake by the host. To address this question, we examined the phagocytosis of the bacterium in differentiated U937 cells after 5 min of challenging. Consistent with previous studies [6, 7, 52], the wild-type *L. pneumophila* strain exhibited a higher

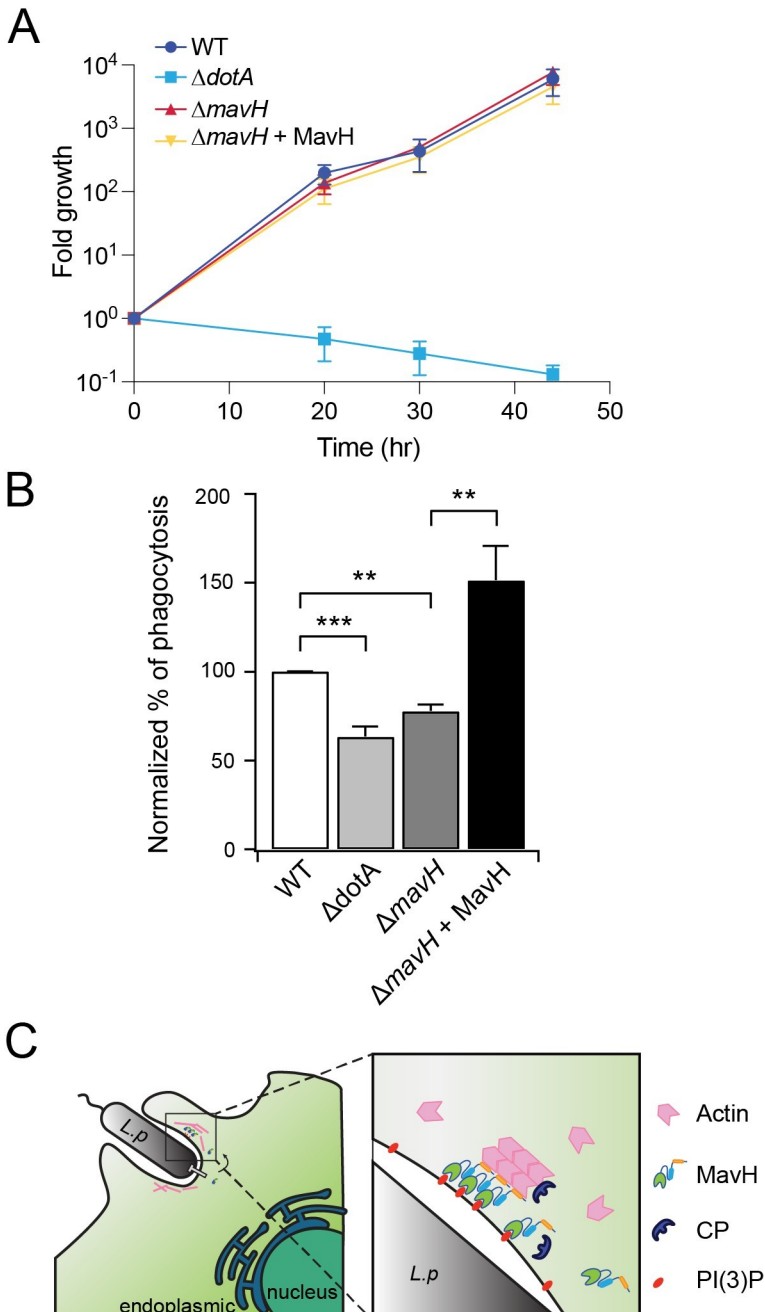

**Fig 8. MavH localizes to the surface of LCV at the early stage of infection.** (A) Intracellular growth assay of *Legionella* in *A. castellanii* host. A wild-type *Legionella* strain, the Dot/Icm deficient *ΔdotA*, the *mavH* deletion strain, and *mavH* deletion strain overexpressing 4xHA-MavH were used to infect *A. castellanii* cells. Growth was assayed by plating colony-forming units (CFUs) at the indicated time after infection. The growth assays were performed in triplicate. (B) (C) A hypothetic model of MavH at the early stage of *Legionella* infection. MavH is secreted at the early stage of infection. Through the PI(3)P interaction, it is clustered at the PI(3)P-membrane-bound organelles to promote actin assembly coupled with membrane remodeling, facilitating the bacterial entry of host cells.

efficiency of internalization by the host than the *ΔdotA* strain, indicating *Legionella* effectors contribute to bacterial entry into the host. Interestingly, the *ΔmavH* strain showed a slight decrease in the phagocytosis rate by U937 cells, whereas the phagocytosis rate was significantly

enhanced when the Δ*mavH* strain was supplemented with a plasmid over-producing MavH (Fig 8B). Together, our data demonstrated that although MavH is nonessential for *Legionella* intracellular growth in *A. castellanii*, it facilitates the entry of the bacterium into the host.

## Discussion

To date, three major classes of eukaryotic nucleators have been identified: the Arp2/3 complex, the formins, and the tandem WH2-domain actin-binding domain proteins. These actin nucleators apply distinct mechanisms for actin nucleation. The Arp2/3 complex is a seven-subunit complex, of which, the Arp2 and Arp3 subunits form a structural mimic of an actin dimer and serve as the nucleator for F-actin assembly. Upon activation by nucleation-promoting factors, the Arp2/3 complex facilitates actin assembly to form a branched actin filament from an existing actin filament [18] or linear actin filaments in the absence of a preformed actin filament [53, 54]. The formins possess characteristic formin-homology 1 domain, which recruits profilin-actin, and formin-homology 2 domain, which mediates the dimerization of formins and facilitates the addition of actin monomers from profilin-actin to the barbed end of the actin filament [19]. The third family of actin nucleators, including Spire [20] and Cobl [21], contain tandem repeats of actin-binding motifs, such as the WH2 domain. These tandem actin-binding domains serve as a scaffold to recruit actin monomers and synergize with other functional domains for F-actin assembly [22]. Interestingly, many bacterial actin nucleators are found to utilize the WH2 domain for actin assembly. For example, the *Vibrio cholerae* and *V. parahaemolyticus* virulent effectors, VopF and VopL, respectively, have three tandem WH2 domains in each chain of a dimer [36, 37, 55], while Arp2/3 activator ActA from *Listeria monocytogens* and Ena/VASP mimic *Burkholderia* BimA both contain two actin binding domains similar to WH2 domains of host proteins [58,][56, 57].

The actin cytoskeleton undergoes dynamic remodeling, which is vital for maintaining cell physiology. Many intracellular bacterial pathogens have evolved distinct strategies to alter host actin cytoskeleton dynamics at different infection stages. These strategies encompass the entry into host cells [58, 59], intracellular movement relying on actin [59, 60], and the formation of the "actin cocoon structure" surrounding the vacuole to evade endocytic degradation [61]. While many pathogenic actin-interacting effectors mimic the host actin assembly proteins (structurally or functionally), some toxins employ unique mechanisms of actin hijacking. The former group can be exemplified by the *Rickettsia* effector Sca2, which promotes filament barbed-end elongation similar to host formins [62, 63]; *Burkholderia* BimA, which utilizes WH2 motifs and poly-proline-rich regions to nucleate, elongate, and bundle actin filaments mimicking host Ena/VASP proteins [56]; *Chlamydia* TARP, which binds to G-actin with its WH2-like domain and, upon oligomerization, nucleates new actin filaments imitating host WH2 domain-containing nucleators [64]. *Vibrio* VopF/L represent bacterial effectors, which employ an intricate combination of a unique pointed-end elongation mechanism [37] and a host-mimicking actin nucleation strategy [36]. An example of an exclusive bacterial effector toxicity mechanism is *Vibrio* and *Aeromonas* actin cross-linking domain (ACD) toxin, which covalently cross-links actin monomers into oligomers with unique properties poisoning several families of host actin assembly proteins [65].

Here we report a novel mechanism of actin polymerization catalyzed by the *Legionella* effector, MavH. Unlike other actin assembly factors, MavH harbors a single actin-binding WH2-like domain and inhibits actin polymerization via a leaky capping activity in the solution. However, binding to PI(3)P-membranes allows the clustering of MavH at the membrane surface (functionally resembling the oligomerization of some other actin nucleators) and promotes robust membrane-bound actin assembly coupled to membrane remodeling. The

membrane tubules induced by MavH are likely stabilized by longitudinally bound F-actin fibers. These observations raise intriguing questions, for example, why does MavH recruit CP while possessing its own capping activity; how does MavH facilitate actin polymerization on membrane surface; and how does MavH-mediated actin polymerization induce membrane deformation and tubulation? Future experiments, such as high-resolution Cryo-EM studies, are needed to address the molecular mechanism of MavH-mediated actin polymerization. Nevertheless, our study uncovers a novel factor that promotes actin assembly in a PI(3)P membrane-dependent manner. Our results may also inspire the discovery of new MavH-like actin polymerization factors in other pathogens or eukaryotes.

MavH is a unique actin assembly promoting factor in that it contains a CPI motif following the actin-binding WH2-like domain. In this study, we showed that MavH recruits CP to endosomes when ectopically expressed in mammalian cells. We found that MavH can modulate actin dynamics and actin density around the liposomes through its recruitment of CP via its CPI motif, but the physiological consequences of this recruitment during infection remain unknown. CPI motif-containing proteins have been shown to recruit CP to specific cellular locations [26] and/or regulate actin-capping activity by allosteric effects [27, 45]. While inhibiting the growth of individual filaments, capping by CP positively contributes to the actin branched network dynamics by i) channeling the limited pool of G-actin to a fewer number of productively growing branches [66] and ii) by releasing nucleation-promoting factors (NPFs) from non-productive association with the barbed ends [67]. Similarly, we speculate that MavH may fine-tune actin dynamics and possibly promote actin network formation, which is important for cellular membrane movement in a number of cellular processes, including phagocytosis. Further experiments are needed to elucidate the biological significance of the recruitment of CP in MavH-mediated actin assembly.

MavH also contains a C-terminal PI(3)P-binding domain [43]. In this study, we characterized the recognition of PI(3)P by MavH and demonstrated that MavH promotes actin polymerization upon its binding to PI(3)P. One possible mechanism for the PI(3)P-dependent actin polymerization mediated by MavH is that the binding of PI(3)P by the C-terminal domain may induce structural rearrangement of the protein, which allows the N-terminal WH2 domain to be fully exposed for actin recruitment and assemble. In this study, we also demonstrated the importance of PI(3)P binding in its anchoring to the LCV during infection. As PI(3)P is tightly regulated by multiple *Legionella* effectors, it accumulates on LCV within 60 s after uptake and is gradually converted into PI(4)P during LCV maturation [68]. Our results, showing that MavH is observed on LCV during the early stage of *Legionella* infection and gradually tapering as the infection progresses, are in agreement with the role of PI(3)P in the early stages of LCV maturation.

In summary, we identified a novel actin assembly factor from *L. pneumophila* that not only triggers actin polymerization in a membrane- and PI(3)P-dependent manner but also mediates membrane remodeling into tubules. We also found that MavH was delivered at the very early stage of infection and localized to the LCV via its binding to PI(3)P, and facilitated the entry of the bacterium into the host. These observations allowed us to propose a functional model for MavH that should be verified and detailed in future studies (Fig 8C). In this model, the translocation of MavH into the host facilitates actin assembly at the phagocytic site, which in turn drives membrane deformation and promotes the internalization of the bacterium. In line with the important role of host actin cytoskeleton dynamics at different infection stages and the necessity of its alteration by intracellular bacterial pathogens, it is interesting to note that several other *Legionella* effectors were found to perturb the dynamics of the host actin cytoskeleton. The *Legionella* effector VipA was shown to promote F-actin assembly and alters host cell membrane trafficking, however, the mechanism for promoting actin polymerization

by VipA is not understood [35, 38]. Another *L. pneumophila* effector, RavK is reported to disrupt actin structures by direct proteolytic cleavage of actin [40]. Two other *Legionella* effectors, LegK2 and WipA, alter the phosphorylation state of the host Arp2/3 complex and inhibit action polymerization [41, 42]. These studies underscore the importance of actin in *Legionella* infection and shed light on the intricate control of the host actin cytoskeleton during the infection process. It will be important for future studies to elucidate how these effectors, which have synergic or antagonistic activities on actin dynamics, orchestrate to exploit the host actin cytoskeleton for successful infection.

## Materials and methods

### Cloning and site-directed mutagenesis

Full-length MavH (a.a. 1–266) was amplified from *L. pneumophila* genomic DNA and digested with BamHI/SalI and inserted into a pET28a-based vector in-frame with an N-terminal His-SUMO tag. PCR products for MavH truncations were amplified from the constructed pET28a His-Sumo MavH. Mutations of MavH were introduced by in vitro site-directed mutagenesis using specific primers containing the defined base changes and PrimeSTAR Max DNA Polymerase (Takara Bio, Inc.) premix. For mammalian expression, corresponding fragments of MavH were subcloned into the pEGFP-C1 vector. For the MavH-SetA chimeric fusion, the N-terminus of MavH (a.a. 13–65) was amplified and digested with BamHI and SalI and then ligated into the pEGFP-C1 vector digested with BglII and SalI. The C-terminal PI(3)P-binding domain of SetA (a.a. 507–629) was amplified and digested with SalI and BamHI and then ligated into pEGFP-MavH (a.a. 13–65) digested with SalI and BamHI to generate pEGFP-MavH-SetA for the expression of the MavH-SetA fusion protein.

Bacterial expression of CP plasmid was purchased from Addgene (Plasmid #89950). The α-subunit of CP was subcloned into a pRSFDuet-based vector in frame with an N terminal His-Sumo tag and the β subunit of CP was subcloned into a pCDFDuet-based vector ORF2. A single residue mutation of CPβ-R15A was introduced by site-directed mutagenesis. For mammalian expression of CP, we constructed pCW57-mCherry-CPβ-P2A-HA-CPα. First, the HA-tagged CP α-subunit was amplified and digested with AvrII and BamHI and cloned into the pCW57-P2A vector to generate pCW57-P2A-HA-CPα. Then mCherry-CPβ was first cloned into the mCherry-C1 vector (restriction sites BglII and SalI) and mCherry CPβ was then amplified and digested with NheI and SalI and inserted into pCW57-P2A-HA-CPα to finally generate pCW57-mCherry-CPβ-P2A-HA-CPα.

For *Legionella* expression, MavH was subcloned into a pZL507-based vector (gift from Dr. Zhao-Qing Luo, Purdue University) with a 4xHA tag. For yeast expression, corresponding fragments of MavH were subcloned into p415gal-yemCherry and p415gal-yeGFP vectors (gift from Dr. Anthony Bretscher, Cornell Univerisity). All constructs were verified by DNA sequencing.

### Protein expression and purification

All MavH constructs in pET28a-His-Sumo were transformed into the Rosetta (DE3) strain of *Escherichia coli* cells using the antibiotic selection markers kanamycin and chloramphenicol. For CP proteins expression, the pRSFDuet HisSumo-CPα subunit and pCDFDuet CPβ subunit were co-transformed into the Rosetta (DE3) strain of *E.coli* cells using the antibiotic selection markers kanamycin and spectinomycin. Transformed bacterial cells were grown in 1L expression cultures at 37°C at 220 rpm and induced with 0.2 mM IPTG during log-phase growth (O.D.$_{600}$ = 0.6–0.8). Cells were incubated at 18°C and 180 rpm for 18 hours post-induction. Cells were collected by pelleting expression cultures at 4000 rpm for 30 minutes at

4˚C. Cells were resuspended in 35 mL of 20 mM Tris (pH 7.5) and 150 mM NaCl containing 1 mM PMSF. Cells were lysed by two rounds of sonication at 50% amplitude, 2-minute duration, and 2 sec on/off pulse on ice. Sonicated samples were spun at 16,000 rpm for 30 minutes at 4˚C to remove the insoluble fraction. The supernatant was collected and mixed with 2 mL of cobalt resin and incubated while rotating for 2 hours at 4˚C to bind proteins. The protein-bound resin was washed with several column volumes of buffer containing 20 mM Tris (pH 7.5) and 150 mM NaCl to remove unbound and nonspecifically bound proteins. The resin was resuspended in 4 mL of wash buffer and cut overnight with His-tagged Ulp1 at 4˚C. Cut proteins were eluted the next day and concentrated to a final volume of 3 mL using a 30 kDa cut-off centrifugal concentrator. Proteins were run on a Superdex200 16/200 column using an AKTA GE Healthcare FPLC system. Peak fractions were collected and analyzed by SDS-PAGE. Purified proteins were further concentrated and stored at -80˚C.

## Cell culture, transfection, fluorescent microscopy and quantification of colocalization

HeLa, Cos7, RAW 264.7, and HEK-293T cells were cultured in Dulbecco's modified minimum Eagle's medium (DMEM) supplemented with 10% FBS fetal bovine/calf serum (FBS). For intracellular localization, EGFP-MavH constructs were expressed in HeLa cells, endosomes were marked by staining of EEA1 via anti-EEA1 rabbit monoclonal primary antibodies and Alexa 647 anti-rabbit secondary antibodies, and actin was stained with Alexa Fluor 488-phalloidin. For colocalization analysis, EGFP-MavH constructs were also co-expressed with RFP-2xFYVE domain constructs in HeLa cells. For CP localization, GFP-tagged MavH constructs were co-expressed with mCherry-CPβ-HA-CPα in HEK293T cells, and 1μg/ml doxycycline was used to induce the expression of CP during transfection.

For imaging, HeLa or HEK293T cells were passaged at 25–30% initial density in a 24-well plate in D10 media. Cells were subsequently transfected 24 hours later with 0.15 μg of each plasmid and a 1:5 (m/v) ratio of polyethyleneimine (PEI) in DMEM for a total volume of 50 μL. At 14–16 hours post-transfection, cells were fixed in 4% paraformaldehyde in PBS solution for 20 minutes on ice and then washed three times with PBS. Fixed coverslips were mounted onto glass slides using Fluoromount-G mounting solution. Fixed cells were imaged using a spinning disk confocal microscope (Intelligent Imaging 108 Innovations, Denver, CO) equipped with a spinning disk confocal unit (Yokogawa CSU-X1), an inverted 109 microscope (Leica DMI6000B), a fiber-optic laser light source, a 100× 1.47NA objective lens, 110 and a Hamamatsu ORCA Flash 4.0 v2+ sCMOS camera. Images were acquired and processed using the Slidebook (version 6) software.

For quantification of colocalization of GFP fusion proteins with EEA1, actin, or RFP-FYVE, Pearson's correlation coefficient (r) was determined using the Coloc2 plugin in ImageJ. Each cell was individually selected for analysis. The resulting data were presented as scatterplots, displaying mean ±SD from a total of 60 cells across three independent experiments. Statistical significance was assessed using one-way ANOVA (****$P < 0.0001$, ***$P < 0.001$, **$P < 0.01$).

## Immunoprecipitation

HEK-293T cells were passaged at 25–30% initial density in a 6-well plate in D10 media. Cells were subsequently transfected with 1.8 μg of each plasmid and a 1:5 (m/v) ratio of polyethyleneimine (PEI) in DMEM for a total volume of 200 μL. At 24 hours post-transfection, cells were washed two times with cold PBS and resuspended in 300 μL of IP lysis buffer (1% Triton-X100, 0.1% deoxycholate in 50 mM Tris, pH 8.0, 150 mM NaCl, and protease inhibitor cocktail (Roche)). Cells were briefly sonicated at 10% amplitude for 5 seconds (pulse) and centrifuged at

15000 rpm for 15 minutes at 4˚C to remove the insoluble fraction. GFP-nanobody conjugated resin was added to the collected supernatant and incubated for 3 hours on a nutating mixer at 4˚C to bind GFP-tagged proteins. Resins with bound proteins were washed with 1 mL of cold PBS for a total of 4 washes. Proteins were eluted from the resin by boiling at 95˚C for 4 minutes in 25 μL of SDS sample loading buffer containing 2% BME. Immunoblotting of GFP-MavH was performed using a homemade rabbit anti-GFP antibody at a dilution of 1:1000. Actin was probed using a mouse anti-Actin antibody (Proteintech) at a dilution of 1:1000. HA tagged CPα was probed using a mouse anti-HA antibody (Sigma). Probed proteins were detected using donkey anti-rabbit IgG antibody, DyLight 800 (Invitrogen), and donkey anti-mouse IgG antibody, Alexa Fluor 680 (Invitrogen; cat. no. A10038) secondary antibodies. Membranes were scanned using a LI-COR Odyssey CLx Imager. Western Blot images were processed and analyzed using ImageStudio Lite software (version 5.2). The samples were subsequently probed with mouse anti-HA (Sigma), rabbit anti-GFP, or mouse anti-actin antibody (Proteintech).

### Liposome preparation

1-palmitoyl-2-oleoyl-sn-glycero-3-phosphocholine (POPC) and 1-palmitoyl-2-oleoyl-sn-glycero-3-phospho-L-serine (POPS) and di-C16-phosphatidylinositol polyphosphates were purchased from Avanti. Liposomes were prepared with POPC and POPS (8:2 molar ratio) or POPC, POPS, and Phosphatidylinositols (8:1:1 molar ratio). Lipid mixtures dissolved in 90% chloroform and 10% methanol were dried in glass tubes by nitrogen gas in the fume hood and rehydrated into G-actin buffer followed by 1 h incubation at 37˚C for the spontaneous formation of liposomes. The liposomes for pyrene-actin polymerization were filtered 10 times through 0.1 μm diameter polycarbonate membranes (Nucleopore).

### Liposome co-sedimentation assay

Purified proteins (1 μM) were incubated with 0.25 mM of liposomes for 20 min at room temperature and then spun down in a benchtop ultracentrifuge (Thermo Fisher AccuSpin Micro 17R centrifuge) for 15 min at 17,000 g. Resuspended pellets and supernatants were analyzed by SDS-PAGE, and quantified by Image J [69]. The fraction of protein in the pellet was calculated as a ratio of protein in the pellet to the total protein amount (pellet + supernatant).

### Pyrene-actin polymerization assay

Actin was purified from muscle acetone powder as described previously [70] followed by gel filtration at low ionic strength to isolate monomeric ATP-G-actin [71]. Purified actin was stored in G-actin buffer (2 mM Tris-HCl (pH 8.0) 0.2 mM $CaCl_2$, 0.2 mM ATP, and 0.1 mM DTT). The actin polymerization biochem kit (BK003) was purchased from Cytoskeleton Inc. Actin polymerization assays were performed in 200 μl reactions using 96-well black polystyrene assay plates. Reactions were started by adding actin to a mix of all other components and 10X actin polymerization buffer (500 mM KCl, 20 mM $MgCl_2$, 0.05 M guanidine carbonate, and 10 mM ATP). Fluorescence was measured in a Tecan Safire2 fluorescence plate reader using excitation/emission wavelength 350 nm (± 20 nm) / 410 nm (± 20 nm). All actin polymerization reactions were performed using 3 μM actin (10% pyrene labeled actin). Liposomes were used at 50 $\mu$M.

### Liposome imaging

Near-infrared DiI dye (Invitrogen) was added when making liposomes to aid the visualization of liposomes. 250 nM wild-type or mutant MavH proteins were incubated with 3μM actin and

250 μM corresponding liposomes at room temperature for 30min. Then polymerization was induced by adding 10X polymerization buffer for 30 min. F-actin was stained by Alexa Fluor 488-phalloidin (Invitrogen) 20 min before imaging. For imaging, 5 μL of the reaction mixture was added to the chamber created between a cover slip and a glass slide. Fluorescence microscopy images were acquired using a spinning disk confocal microscope (Intelligent Imaging 108 Innovations, Denver, CO) equipped with a spinning disk confocal unit (Yokogawa CSU-X1), an inverted 109 microscope (Leica DMI6000B), a fiber-optic laser light source, a 63X, and a 40X objective lenses. Images were acquired and processed using the Slidebook (version 6) software.

## In vitro reconstituted TIRFM assays

TIRFM experiments were conducted essentially as described [37] using 1.5 μM 20% Alexa Fluor 488–labeled actin (final concentration) in protocatechuate dioxygenase (PCD)/protocatechuic acid (PCA) oxygen scavenging system [72]. To measure actin nucleation and filament elongation, actin was mixed with 250 nM MavH and 50 μM liposomal lipids (prepared as described above and labeled with DiD perchlorate (AAT Bioquest, Pleasanton, CA)) in the final TIRF buffer (pH 7.0): 10 mM imidazole, 0.2 mM EGTA, 1 mM MgCl2, 50 mM KCl, 0.25 mM ATP, 10 mM ascorbic acid, 2.5 mM PCA (MilliporeSigma, Burlington, MA), 0.1% bovine serum albumin (VWR International, Radnor, PA), 0.6% methylcellulose-400cP (MilliporeSigma, Burlington, MA), and 0.1 μM PCD purified as described [37]. Immediately upon mixing, the timer was set to account for the total polymerization time, and the reactions were transferred to a TIRF chamber. Time-lapsed imaging in four fields of view simultaneously was started within the first 1.5 min of actin polymerization. Actin nucleation was assessed by counting the number of filaments present in a field of view at a 4-min time point (post-mixing). Filament elongation was measured by plotting the length of individual actin filaments over time. Kymograph images of actin filaments were obtained using ImageJ [69].

To investigate membrane remodeling, liposomes in the TIRF buffer were allowed to attach to the glass surface in a TIRF chamber for 5 min, the time-lapsed imaging was started, after which the actin with MavH in the TIRF buffer was flowed in immediately upon their mixing without pausing the imaging. This procedure allowed for better detection of the membrane elongation/tubulation process, which was obscured in alternative experimental designs. The images were acquired using a Nikon Eclipse Ti-E microscope equipped with a TIRF illumination module (with a 15-mW laser), a Nikon CFI Plan Apochromat λ 100× oil objective (NA: 1.45), a perfect focus system (Nikon Instruments, Melville, NY), and an iXon Ultra 897 EMCCD camera (Andor Technology, Belfast, UK).

## Sample preparation and image acquisition for negative stain EM

PI(3)P-containing liposomes underwent 5 rounds of freeze-thawing process in liquid nitrogen to make small liposomes. Samples were prepared by incubating 1 μM MavH, 6 μM actin, and 500 μM liposomes for 30 min, and actin polymerization was induced by adding 10X polymerization buffer for 30min. Carbon-coated EM grids (200 mesh, from Electron Microscopy sciences) were glow-discharged using PELCO easiGlow Glow Discharge Cleaning system. A sample of 5 μl was applied to the grids, followed by incubation for 1 min, and the excess sample was absorbed by the paper. Then 2% uranyl acetate was used to stain for 1min followed by absorption of the excess stain with paper. Negative stain images were obtained using an F200C microscope (Thermo Scientific Talos).

## EGF trafficking assay

COS-7 cells were split and cultured on poly-lysine-coated cover glass in a 24-well plate. Cells were transfected with the plasmid of EGFP or EGFP-tagged MavH for 24 hours. Cells were incubated with 20 ng/mL Alexa 555-EGF (in DMEM) on ice for 20 min, washed using ice-cold DMEM three times, and then incubated at 37˚C in a $CO_2$ incubator for the indicated time. After EGF uptake, cells were fixed using 4% paraformaldehyde (PFA), permeabilized with 0.1% saponin. To label early endosomes, cells were immunostained with rabbit anti-EEA1 primary antibodies and then with Alexa 647 anti-rabbit secondary antibodies.

## Yeast strains and growth assays

For MavH localization in yeast cells, the SEY6210.1 yeast strain expressing genomically tagged VPH1-mCherry is obtained from Dr. Scott Emr (Cornell Univ.). This strain was transformed with the plasmid pRS416 Gal3, together with the pRS415 plasmid expressing either GFP-MavH wild type or mutants under a galactose-inducible promoter and was grown overnight in a complete supplement mixture -Ura-Leu media (Sunrise Science Products) containing 2% glucose at 30˚C. Cultures were then centrifuged at 1000 xg and pellets were resuspended in selective media containing 2% galactose and incubated for 4 hours at 30˚C with agitation to induce protein expression. For the effects of MavH on the actin cytoskeleton in yeast cells, the BY4741 strain expressing either mCherry-MavH wild-type, truncations, mutants, or with vector controls under galactose inducible promoters were grown overnight in a complete supplement mixture -Leu media containing 2% glucose at 30˚C. Cultures were then centrifuged at 500 g and pellets were resuspended in selective media containing 2% galactose and incubated for 4 hours at 30˚C with agitation to induce protein expression. In 3 ml cell cultures, 750 µl of 20% PFA was added and incubated for 30min to fix cells and then cells were washed three times with PBS. PBS containing 0.2% Triton-X100 was used to permeabilize cells and Alexa Fluor 488-phalloidin (Invitrogen, A12379) was used for actin staining. Cells were immobilized on coverslips using concanavalin A and imaged using a spinning disk confocal microscope. Yeast growth assays were performed as described [73]. Briefly, yeast cultures for each strain were grown to log phase and diluted to $OD_{600}$ = 0.3 and then serially diluted by a factor of 10. 5 µL of each serially diluted sample was either spotted on -Leu plates containing 2% glucose for repression or 2% galactose for expression of proteins and then incubated for 2–3 days at 30˚C before imaging.

## *Legionella* strains and infection

Strains of *L. pneumophila* used were the wild-type Lp02 and the Dot/Icm deficient Lp03 strain (*ΔdotA*) [74]. MavH deletion strain was created by a two-step allelic exchange strategy as described [75]. Briefly, a 1.2kb DNA fragment from upstream and downstream of MavH was amplified by PCR including the first 15 amino acids and the last 14 amino acids of the gene. The SalI and BamHI restriction sites were introduced into the 5' and 3' end of the upstream fragment while the BamHI and SacI sites were introduced into the 5' and 3' end of the downstream fragment, respectively. These fragments digested with the proper enzymes are ligated into plasmid pSR47S and digested with SalI and SacI by three-way ligation. Plasmids were introduced into *L. pneumophila* by conjugation with *E.coli* donors as described previously [76]. Conjugants containing plasmids integrated into bacterial genome were selected on CYET plates containing kanamycin (20 µg/ml) and streptomycin (50 µg/ml). Kanamycin-resistant *L. pneumophila* colonies were plated on CYET plates containing 5% sucrose to allow selection against the plasmid that contains the sacB gene. MavH deletion strains were verified by PCR. MavH complementary strains were generated by transformation using 4x-HA-tagged mavH

or its mutants in the pZL507 vector. 4xHA-MavH expression was induced with 0.1 mM IPTG for 30 min before infection.

To detect MavH localization or F-actin after *Legionella* infection, HEK-293T cells and U937 cells were used as the host of bacteria. For HEK-293T cell infection, cells were transfected with FcγRII for 24 hrs, and bacteria of indicated *Legionella* strains were opsonized with rabbit anti-*Legionella* antibodies (1:500) at 37°C for 20 min before infection. For U937 cell infection, cells were induced with 10 ng/mL phorbol 12-myristate 13-acetate (PMA) for two days before infection. FcγRII-expressing HEK-293T cells or PMA-induced U937 cells were infected with post-exponential *L. pneumophila* strains at an MOI of 5 for the indicated amount of time. Cells were then fixed using 4% PFA for 15 min. To detect MavH localization in HEK-293T cells, fixed cells were permeabilized using ice-cold methanol for 10 min. 4xHA-MavH was immunostained using mouse anti-HA primary antibodies (1:1000) and Alexa 488 anti-mouse secondary antibodies. To detect F-actin in host cells, fixed cells were permeabilized with 0.1% Triton X-100 for 30 min. F-actin was stained with Alexa Fluor 488- or Rhodamine-conjugated phalloidin (1:100 for both) for 2 hours.

## Bacterial growth assays

The intracellular growth assay of *L. pneumophila* was assayed as described previously [77]. Briefly, *A. castellanii* was propagated using PYG medium. Cells were grown to near confluency and then plated into 24-well plates at a density of one million per well, and then infected with stationary phase *L. pneumophila* at an MOI of 0.3 for 1 hr. The cells were then washed one time to remove extracellular bacteria and incubated at 37°C for the indicated period. The amoeba cells were lysed with 0.05% saponin (Alfa Aesar, A18820) in PBS at 0, 20, 30, and 44 h, and the serial dilutions of lysates were plated with serial dilutions onto CYE agar plates. Bacterial colonies were counted after 4 days of incubation at 37°C. All growth assays were performed in triplicate.

## Bacterial phagocytosis assay

Differentiated U937 cells were challenged with indicated *L. pneumophila* strains at an MOI of 100 for 5 min. Cells were then fixed using 4% PFA for 15 min. To stain host extracellular *Legionella*, U937 cells were incubated with rabbit anti-*Legionella* primary antibody and then with Alexa 568 anti-rabbit secondary antibody. After that, to stain total *Legionella*, cells were permeabilized with 0.1% Triton X-100, then incubated with rabbit anti-*Legionella* primary antibody followed by incubation of Alexa 488 anti-rabbit secondary antibody. Cells were imaged using confocal fluorescence microscopy, and U937 intracellular *Legionella* was determined by counting the number of bacterial cells negative for Alexa 568 and positive for Alexa 488. The average numbers of bacteria per U937 cell for all *Legionella* strains were calculated and the values for other *Legionella* strains were further normalized by the wild-type Lp02 strain.

## Supporting information

**S1 Fig. MavH co-localizes with the RFP-FYVE motif.** Hela cells were co-transfected with a plasmid expressing RFP-2xFYVE and either with GFP, GFP-MavH, or GFP-MavH R162A/H163A for 20 hours. The colocalization of MavH with the PI(3)P marker RFP-2xFYVE was analyzed by confocal fluorescence microscopy. Wild-type MavH showed a high degree of colocalization with RFP-2xFYVE, while the MavH R162A/H163A mutant defective of PI(3)P-binding exhibited no colocalization with RFP-2xFYVE. Scale bars, 10 $\mu$m. (B) Colocalization of GFP fusion proteins with RFP-FYVE. Colocalization was determined by Pearson's correlation coefficient r. Data are depicted as scatterplots showing mean ±SD from 60 cells in 3

independent experiments. Statistical significance was assessed using one-way ANOVA (****P<0.0001).
(TIF)

**S2 Fig. MavH inhibits endosomal trafficking.** Cos7 cells were first transfected with indicated plasmids for 24 hours. Cells were then incubated with 20 ng/mL Alexa 555-EGF on ice for 20 min, washed, and then incubated at 37˚C for the indicated time. Early endosomes were stained with EEA1 antibodies. Representative images are shown for EGFP (A) or EGFP-MavH (B) transfected cells. EGFP-tagged protein is shown in green, Alexa 555-EGF is in red, and EEA1 is in blue. Scale bars are 10 $\mu$m. (C) EEA1-positive EGF fluorescence was quantified, shown as Mean ± SEM from three independent experiments. At least 28 cells/conditions were analyzed. ***P<0.001, N.S., not significant.
(TIF)

**S3 Fig. MavH binds PI(3)P via the C-terminal domain.** (A) Schematic representation of MavH. N-terminal α-helix (α1) is colored in yellow. The C-terminal domain (CTD) is in green. (B) Ribbon diagram of predicated MavH structure with AlphaFold2. (C) Liposome co-sedimentation assays of MavH. Liposomes were formed with PC, PS, and indicated phosphoinositides. After incubation with MavH, the liposomes were pelleted by ultracentrifugation. P, pellet; S, supernatant. Pellet and supernatant fractions were then analyzed by SDS-PAGE, followed by Coomassie staining. (D) Quantification of the liposome sedimentation assays in (C). The protein factions in the pellet are shown as mean ± SEM from three independent experiments. (E) Molecular surface of CTD of MavH. The surface is colored based on the electrostatic potential with the positively charged region in blue and the negatively charged surface in red. (F) Ribbon representation of CTD of MavH. The conserved positively charged residues, R162 and H163, are shown in sticks. (G) Liposome co-sedimentation assays of MavH wild-type and R162A/H163A mutant. (H) Quantification of liposome sedimentation assays in (G). The protein fractions in the pellet are shown as mean ± SEM from three independent experiments.
(TIF)

**S4 Fig. A chimeric fusion of MavH with SetA_CTD promotes actin polymerization around endosomes.** HeLa cells were transfected with GFP-tagged constructs of either SetA PI(3)P-binding domain (SetA_CTD) alone or a chimeric fusion (MavH-SetA) containing the N-terminal portion of MavH in-frame with the SetA_CTD. After transfection with indicated plasmids for 18 h, cells were fixed and stained with Rhodamine conjugated phalloidin and imaged by confocal microscopy. Scale bars are 10 $\mu$m.
(TIF)

**S5 Fig. Intracellular localization and function of MavH in *S. cerevisiae*.** (A) SEY6210.1 yeast strains expressing integrated copies of mCherry-tagged yeast vacuolar marker VPH1 were transformed with either GFP-tagged wild-type MavH, truncations, or mutant constructs under the control of a galactose inducible promoter. Cells were visualized by fluorescence confocal microscopy after induction of protein expression by selective media containing 2% galactose. (B) BY4741 yeast strains were transformed with either mCherry-tagged wild-type or mutant MavH constructs under the control of a galactose inducible promoter. After induction of protein expression by selective media containing 2% galactose, yeast cells were stained with Alexa Fluor 488-phalloidin and visualized by fluorescence confocal microscopy. (C) Yeast cultures from (B) were grown on plates containing glucose or galactose (inducing conditions). 10-fold serial dilutions of each yeast cell culture were spotted on the plate and the lethal effects

were compared to the empty vector control.
(TIF)

**S6 Fig. MavH promotes actin assembly on PI(3)P containing liposomes.** In vitro liposome imaging assay. Reactions were performed using 3 μM actin and 250 nM MavH. Liposomes were used at 500 μM. After induction of actin polymerization, actin was stained with Alexa Fluor 488-phalloidin. Images were taken by confocal microscopy.
(TIF)

**S7 Fig. MavH localizes to the surface of LCV and facilitates LCV accumulation of F-actin at the early stage of *Legionella* infection in U937 cells.** (A) Representative images show localization of 4xHA-MavH at 5 min post-infection. PMA-induced U937 cells were challenged by wild-type *ΔmavH* or *ΔdotA* strains supplemented with a plasmid expressing 4xHA-MavH for 5 min. Cells were fixed using 4% PFA for 15 min and then permeabilized using ice-cold methanol for 10 min. 4xHA-MavH was immunostained using mouse-anti-HA primary antibodies and Alexa-488 anti-mouse secondary antibodies. Scale bars are 10 μm. (B) Quantification of LCVs positive for 4xHA-MavH in U937 cells infected by WT*ΔmavH* overexpressing 4xHA-MavH for the indicated time, shown as mean ± SEM from three independent experiments. At least 50 LCVs were analyzed for each time point. (C) Representative images show *Legionella* bacteria (green) and Rhodanmine-phalloidin-labeled F-actin (red) in PMA-induced U937 cells challenged by the indicated *Legionella* strains for 10 min. Scale bars are 10 μm. (D) Quantification of LCVs positive for F-actin in U937 cells in (E) shown as mean ± SEM from three independent experiments. At least 50 LCVs were analyzed for each time point. *P < 0.05, N.S. = not significant.
(TIF)

**S1 Video. PI(3)P-membrane deformation by MavH-induced actin polymerization in the *in vitro* TIRFM reconstituted assay.** DiD-labeled PI(3)P-containing liposomes (50 μM) were allowed to attach to the glass surface for 5 min. The time-lapsed imaging was started (-100 s time point), and the mixture of 1.5 μM 20% Alexa Fluor 488-labeled actin with 250 nM MavH was flowed into the TIRF chamber containing the liposomes (0 s time point). Actin is shown in cyan; liposomes are in yellow.
(AVI)

**S2 Video. Control TIRFM experiment conducted in the absence of PI(3)P.** DiD-labeled liposomes (50 μM) prepared without PI(3)P were allowed to attach to the glass surface for 5 min. The time-lapsed imaging was started (-100 s time point), and the mixture of 1.5 μM 20% Alexa Fluor 488-labeled actin with 250 nM MavH was flowed into the TIRF chamber containing the liposomes (0 s time point). Actin is shown in cyan; liposomes are in yellow.
(AVI)

## Acknowledgments

We thank Dr. Anthony Bretscher (Cornell Univ.) for providing beef skeletal muscle acetone powder for actin purification and plasmids (pRS415 yemCherry and pRS415 yeGFP); Dr. Scott Emr (Cornell Univ.) for yeast strain (SEY6210.1 with genomically tagged VPH1-mCherry). We also thank Dr. Joseph Vogel (Wash. Univ.) for the critical discussion.

## Author Contributions

**Conceptualization:** Qing Zhang, Min Wan, Elena Kudryashova, Dmitri S. Kudryashov, Yuxin Mao.

**Data curation:** Qing Zhang, Min Wan, Elena Kudryashova, Dmitri S. Kudryashov.

**Formal analysis:** Qing Zhang, Min Wan, Elena Kudryashova, Dmitri S. Kudryashov, Yuxin Mao.

**Funding acquisition:** Yuxin Mao.

**Investigation:** Qing Zhang, Min Wan, Elena Kudryashova.

**Methodology:** Qing Zhang, Min Wan, Elena Kudryashova.

**Project administration:** Dmitri S. Kudryashov, Yuxin Mao.

**Resources:** Yuxin Mao.

**Supervision:** Yuxin Mao.

**Validation:** Qing Zhang, Min Wan, Elena Kudryashova, Dmitri S. Kudryashov, Yuxin Mao.

**Visualization:** Qing Zhang, Min Wan, Elena Kudryashova, Dmitri S. Kudryashov, Yuxin Mao.

**Writing – original draft:** Qing Zhang, Min Wan, Yuxin Mao.

**Writing – review & editing:** Qing Zhang, Min Wan, Elena Kudryashova, Dmitri S. Kudryashov, Yuxin Mao.

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
