## [Decision Letter · Decision Letter 0]

27 Mar 2023

Dear Dr. Mao,

Thank you very much for submitting your manuscript "Membrane-dependent actin polymerization mediated by the Legionella pneumophila effector protein MavH" for consideration at PLOS Pathogens. As with all papers reviewed by the journal, your manuscript was reviewed by members of the editorial board and by several independent reviewers. In light of the reviews (below this email), we would like to invite the resubmission of a significantly-revised version that takes into account the reviewers' comments.

All the reviewers highly evaluated the biochemical and cell biological characterization of MavH regarding its regulation of actin polymerization associated with PI(3)P-containing liposomes. I agree that the presented data and organization of the manuscript are high quality. The weakness of the study is that the biological implication on Legionella infection is limited, as all the reviewers pointed. Any of the infection experiments the reviewers suggested should be added to Figure 6 to examine a role of MavH in infected cells. I just wonder if the early-endosome traffic to a T4SS-deficient LCV can be reduced (or delayed) by the ectopic expression of MavH. This could be an alternative analysis if the suggested experiments wouldn’t work.

We cannot make any decision about publication until we have seen the revised manuscript and your response to the reviewers' comments. Your revised manuscript is also likely to be sent to reviewers for further evaluation.

Sincerely,

Tomoko Kubori, Ph.D.

Academic Editor

PLOS Pathogens

David Skurnik

Section Editor

PLOS Pathogens

Kasturi Haldar

Editor-in-Chief

PLOS Pathogens

orcid.org/0000-0001-5065-158X

Michael Malim

Editor-in-Chief

PLOS Pathogens

orcid.org/0000-0002-7699-2064

All the reviewers highly evaluated the biochemical and cell biological characterization of MavH regarding its regulation of actin polymerization associated with PI(3)P-containing liposomes. I agree that the presented data and organization of the manuscript are high quality. The weakness of the study is that the biological implication on Legionella infection is limited, as all the reviewers pointed. Any of the infection experiments the reviewers suggested should be added to Figure 6 to examine a role of MavH in infected cells. I just wonder if the early-endosome traffic to a T4SS-deficient LCV can be reduced (or delayed) by the ectopic expression of MavH. This could be an alternative analysis if the suggested experiments wouldn’t work.

Reviewer's Responses to Questions

**Part I - Summary**

Reviewer #1: The study by Zhang et al provides some compelling results to demonstrate that the Legionella pneumophila effector MavH modulates the cytoskeleton of host cells by interacting with capping proteins, promoting actin polymerization in a membrane- (liposome) dependent manner. The study is comprehensive and with a few exceptions, the results are of high quality. The study needs further experiments to address some of the pressing questions in the study of Legionella effectors such as functional redundancy among effectors with similar cell biological impact.

Reviewer #2: The study by Zhang et al. investigates the mode of action of the type IV secretion system (T4SS)-translocated effector protein MavH, produced by the causative agent of Legionnaires’ disease, Legionella pneumophila. MavH was identified in a screen for L. pneumophila effectors that perturb the host actin cytoskeleton. Ectopically produced GFP-MavH showed a punctate localization and co-localized with the early endosomal marker EEA1, in agreement with an earlier study documenting the direct binding of MavH to the endosomal phosphoinositide (PI) lipid PI(3)P. The authors then go on to show that (i) MavH recruits host actin capping proteins (CP) and actin to endosomes via its central CP-interacting (CPI) motif and an N-terminal WASP homology (WH2)-like actin binding domain, (ii) MavH inhibits actin polymerization in absence of membranes in vitro, but stimulates actin polymerization in presence of PI(3)P-containing membranes, (iii) MavN recruits CP to negatively regulate actin density at membranes, and (iv) MavN is secreted by the T4SS and localizes to the pathogen vacuole membrane very early during infection.

The manuscript is well-written, and the story unfolds in a straightforward, logical manner. The putative CPI motif and the WH2 domain of MavH were validated by convincing mutagenesis analysis and conclusive cell biological as well as biochemical readouts. Overall, the finding that MavH adopts a novel mechanism to nucleate actin is very intriguing. In contrast to the known actin nucleation mechanisms (Arp2/3 complex, formins, tandem actin-binding domain proteins), a single actin-binding WH2 domain, in concert with PI(3)P-containing membranes, suffices for the effector MavH to nucleate actin. A few rather minor points should be addressed to further strengthen the study.

Reviewer #3: The manuscript describes a Legionella effector protein, MavH, that has been characterized in the study by the authors to localize to endosomes via a PI3-binding domain. Furthermore, two N-terminal domains were identified that indicate a potential role of MavH for actin polymerization around endosomes during Legionella infection. These domains are a capping-binding protein binding domain, and also a WH2-like domain for actin binding. In vitro assays were performed that demonstrate actin polymerization dependent on these domains in the presence of PI3P-containing liposomes. Also, pulldowns were performed to characterize the interaction with the proposed partners. These in vitro assays are very convincing. Nevertheless, the cell biological characterization through co-localization is very preliminary, the EM data not interpretable, and the infection assays in figure 6 also rather preliminary. Therefore, I feel that the manuscript has too many “open ends” to allow the conclusions of the authors. Even though well written, a huge amount of additional experiments, analysis and controls would need to be performed to link the “in vitro” data with the proposed role of MavH during infection.

**Part II – Major Issues: Key Experiments Required for Acceptance**

Reviewer #1: 1. The regulation of actin dynamics by MavH appears to be complex in which the CPI motif seems to downregulate actin polymerization by recruiting CP proteins whereas its N-terminal region promotes actin polymerization in the presence of liposome (membrane). The effects on actin polymerization shown in Figure 4 are very modest. Can this be due to the presence of the CPI domain? Although this experiment was done in the absence of CPs, the authors may add experiments using MavH mutants lacking the CPI motif. In the same line of thinking, binding to lipid may expose the structural features in MavH directly involved in promoting actin polymerization. This point should be discussed.

2. Based on the results, what is the net outcome of the effect of MavH on the accumulation of actin polymers on the LCV? Inhibition or induction? An inhibitory effect would suggest functional redundancy with the activity of RavK, which raised a questions whether a double mutant lacking both mavH and ravK will be detectably defective in intracellular growth. The authors are in a perfect position to examine this hypothesis.

3. Figure 6, the use of the 4xHA tagged MavH to examine the dynamics of LCV association by this protein is problematic due to the potential infection phase-dependent regulation of effector expression and translocation. These results should be done with a strain in which the tagged version is knocked into the chromosome. Nevertheless, the observation that even the protein is expressed by an artificial promoter on a multicopy plasmid, such association still occurred, suggesting the existence of a mechanism that function to remove MavH from the LCV as infection proceeds. This point should be discussed.

Reviewer #2: 1) Given that MavH is accumulating on the pathogen vacuole membrane at very early time points post infection (Fig. 6B), and an L. pneumophila mutant lacking MavH does not show an intracellular replication phenotype (Fig. 6E), possible effects of MavH on initial bacterial uptake should be assessed. E.g., determine uptake efficiency with fluorescently labelled wild-type and mavH mutant L. pneumophila by FACS.

Reviewer #3: - None of the data on the fluorescence localization data and the co-localizations has been quantified. Nevertheless, this is crucial for this particular study as there is no data presented that co-localization takes place between MavH, the endosomes AND actin patches (three factor co-localizations). Only two factors are shown to co-localize, and never quantified. It is possible that there is no co-localization between the three factors (endosomes, MavH, and actin patches), for example in some cases MavH could co-localize with the actin patches, and in other cases with endosomes. This would need to be addressed systematically throughout the whole manuscript (especially important for fig 1, 2, 3 and 6, and also the suppl figures).

- There is only one figure on Legionella infection (Fig 6), and the chosen host cell type is not the most relevant- it is the HEK293T cell model. All other figures are also in cell types that are not necessarily the best models for Legionella infections. It is possible that there are cell-type specific differences how MavH interacts with different compartments, and this should be addressed in cell models relevant for Legionella infections.

- The fluorescence data in figure 6 does not provide sufficient information on the action of MavH as analyzed in the figures before… are there actin patches somewhere? Where does MavH reprogram host actin? All of this should be analyzed in more detail. Also, a time course would be helpful to understand when MavH host cell reprogramming takes place.

- The data presented in Fig 4B and C does not fit well together. The effects are gradual in Fig 4B, however they appear “black or white” in Fig 4C. One would respect also gradual differences there.

- The EM data lacks any controls and is not interpretable as such.

**Part III – Minor Issues: Editorial and Data Presentation Modifications**

Reviewer #1: 1. Line 28, from the results in Figure 4, the induction of actin polymerization by MavH even in the presence of liposome is not robust at all.

2. Lines 47-48, precisely speaking the two references (Hilbi et al., 2001 and Watarai et al., 2001) did NOT show the role of Dot/Icm effectors in bacterial entry nor did the paper of Roy et al., 1998 (the first effector RalF was published in 2004). The authors should cite some more recent and relevant works. There are many to choose from.

3. Line 166 will MavHΔN93 be better?

4. Line 178, please specify which mutants.

Reviewer #2: 1) Fig. S1: The co-localization of MavH and RFP-FYVE should be quantified. E.g., calculate Pearson correlation coefficients.

2) Abstract: the statement “In this study, we report that the Legionella MavH effector harbors a lipid binding domain …” should be phrased more precisely, as in fact, a previous study already showed that MavH binds to PI(3)P (Nachmias et al., 2019), and in the current study, no biochemical PI lipid binding assays were performed.

3) The concept of “capping activity” is introduced without further explanation (l. 260), and Fig. 5C shows an in vitro actin polymerization assay, rather than a “capping assay”. This should be rephrased and/or explained for the non-specialist reader.

4) Fig. 6B: The kinetics of MavH recruitment to the pathogen vacuole membrane formed in HEK293T cells is very fast (minutes). Were the bacteria spun onto the cells, or just left to infect the cells without centrifugation? Were other, more natural host cells tested (amoeba, macrophages)?

5) Typos:

- l. 64: … Spire (Kerkhoff, 2006) and Cobl (Ahuja et al., 2007) …

- l. 97: … MavH, which localizes …

- l. 195: … domain from the Legionella effector SetA, …

- l. 777: … dependent on the CPI motif.

- Some references are incomplete/incorrect (e.g., l. 701, 706, 711, 714, 742) and not uniformly formatted.

- Fig. 6B (X-axis: L. pneumophila), Fig. 6E (X-axis: Time (hr)).

Reviewer #3: Minor points:

- In figure 2C, the band at around 40kDa should be explained.

- The IPs lack a control for the overall protein input (could be a house keeping protein).

- Fig 4B: why is the MavH signal so jumpy?

PLOS authors have the option to publish the peer review history of their article (what does this mean?). If published, this will include your full peer review and any attached files.

Reviewer #1: No

Reviewer #2: No

Reviewer #3: No
---

## [Decision Letter · Decision Letter 1]

26 Jun 2023

Dear Dr. Mao,

We are pleased to inform you that your manuscript 'Membrane-dependent actin polymerization mediated by the Legionella pneumophila effector protein MavH' has been provisionally accepted for publication in PLOS Pathogens.

Best regards,

Tomoko Kubori, Ph.D.

Academic Editor

PLOS Pathogens

David Skurnik

Section Editor

PLOS Pathogens

Kasturi Haldar

Editor-in-Chief

PLOS Pathogens

orcid.org/0000-0001-5065-158X

Michael Malim

Editor-in-Chief

PLOS Pathogens

orcid.org/0000-0002-7699-2064

I am very pleased to inform you that all the reviewers are fully satisfied with the revised manuscript.

Reviewer Comments (if any, and for reference):

Reviewer's Responses to Questions

**Part I - Summary**

Reviewer #1: The authors have adequately addressed my concerns, and i have no further comments on this submission.

Reviewer #2: The authors did a thorough and convincing job revising the manuscript and addressing the reviewers’ points in a satisfactory manner.

Reviewer #3: I have to say that I have been really impressed how serious the authors have taken this revision and how well they have addressed my points. I thank them for taking them. In summary, I find that the revised version is excellent and an important contribution to the field.

**Part II – Major Issues: Key Experiments Required for Acceptance**

Reviewer #1: none

Reviewer #2: None.

Reviewer #3: The authors have addressed my points extremely well. I feel that the current version is really excellent.

**Part III – Minor Issues: Editorial and Data Presentation Modifications**

Reviewer #1: none

Reviewer #2: None.

Reviewer #3: No minor issues

PLOS authors have the option to publish the peer review history of their article (what does this mean?). If published, this will include your full peer review and any attached files.

Reviewer #1: No

Reviewer #2: No

Reviewer #3: No

---

## [Editor Report · Acceptance letter]

13 Jul 2023

Dear Dr. Mao,

We are delighted to inform you that your manuscript, "Membrane-dependent actin polymerization mediated by the *Legionella pneumophila* effector protein MavH," has been formally accepted for publication in PLOS Pathogens.

Best regards,

Kasturi Haldar

Editor-in-Chief

PLOS Pathogens

orcid.org/0000-0001-5065-158X

Michael Malim

Editor-in-Chief

PLOS Pathogens

orcid.org/0000-0002-7699-2064